# Learning Deep Modality-Shared Self-Expressiveness for Image Clustering with Textual Information

## Abstract

Leveraging textual information for image clustering has emerged as a promising direction, driven by the powerful representations of vision-language models. However, existing approaches usually leverage modality alignment, which merely shapes the representations implicitly, failing to preserve and exploit modality-specific structures, and leaving the overall representation distribution unclear. In this paper, we propose a simple but principled approach, termed deep **mo**dality-sha**r**ed **s**elf-**e**xpressive model (DeepMORSE), which simultaneously learns structured representations that conform to the union of modality-specific subspace structures and, via a modality-shared self-expressive model, discovers structures shared across modalities. We evaluate our DeepMORSE approach on seven widely used image clustering benchmarks and observe performance improvements exceeding 4% on the UCF-101, DTD-47, and ImageNet-Dogs datasets. In addition, we demonstrate the strong transferability of the learned representations by achieving state-of-the-art performance on downstream tasks such as image retrieval and zero-shot classification——without requiring any task-specific losses or post-processing.

## 1 Introduction

Clustering is a fundamental unsupervised learning task that aims to partition data according to its latent patterns or distributions without relying on ground-truth labels (Forgy, 1965; MacQueen, 1967). Although classical clustering algorithms perform well on simple datasets (e.g., MNIST Le-Cun et al. 1998), they often struggle with high-dimensional, real-world images that exhibit complex structures (Krizhevsky et al., 2009; Deng et al., 2009). Over the past decade, deep clustering has emerged as a powerful approach by leveraging the representation learning capabilities of neural networks (Lu et al., 2024; Ren et al., 2025) and progressed rapidly following the advances in representation learning——from early autoencoder-based approaches (Xie et al., 2016; Guo et al., 2017; Yang et al., 2017) to contrastive learning-based approaches (Van Gansbeke et al., 2020; Li et al., 2021; Zhong et al., 2021; Dang et al., 2021), and more recently to the pre-trained (multimodal) representation-based approaches (Adaloglou et al., 2023; Cai et al., 2023; Chu et al., 2024; Li et al., 2024).

Since the essence of clustering lies in discovering the underlying patterns of the data, an important question in multimodal clustering is: *what kinds of patterns should we aim to capture?* In existing clustering methods, the underlying patterns of the data are often modeled as, e.g., centroids (Forgy, 1965; MacQueen, 1967; Arthur & Vassilvitskii, 2007), low dimensional (linear or affine) subspaces (Vidal, 2011; Vidal et al., 2016), or low dimensional manifolds (Patel & Vidal, 2014; Elhamifar & Vidal, 2011; Li et al., 2022; Ding et al., 2023). While such a geometric perspective applies to classical clustering, when dealing with multimodal data, we have to consider how to effectively leverage information from different modalities. For example, although images and their textual counterparts share the same semantics, they represent the semantics in different forms and thus have the specific limitations: images contain rich pixel-level details but are sensitive to noise from background or occlusion, whereas textual descriptions provide semantic concepts but might occur mismatches (due to the retrieval process). We argue: *the most reliable underlying patterns in*

Figure 1: **Illustration of the basic idea of the paper.** Existing approaches typically match pseudo-labels of multimodal data, leaving the geometric structure of the learned multimodal representations unclear. In contrast, we explicitly learn representations that conform to a union of modality-specific subspaces, and clustering by capturing modality-shared structure.

*multimodal data reside in the low dimensional structures shared across different modalities*—which is essentially the *"wisdom of the crowd"*.

Recently, the pre-trained vision-language models are leveraged for image clustering (Adaloglou et al., 2023; Chu et al., 2024; Meng et al., 2025), but the textual data are not explicitly incorporated into the clustering algorithms, thus these approaches are essentially still unimodal clustering methods. In contrast, (Cai et al., 2023; Li et al., 2024) involve textual data explicitly into image clustering by aligning pseudo-labels across modalities and neighbors; however, without explicitly modeling the geometric structures of data in each modality (e.g., as a mixture of centroids, subspaces, or manifolds), performing alignment across different modalities may disrupt the internal distribution of each modality and make the aligned geometric structure of the learned representations unclear.

In this paper, we propose a deep **mo**dality-sha**r**ed **s**elf-**e**xpressive model (DeepMORSE) that simultaneously learns modality-specific representations that conform to desired structures and captures shared structures across modalities. Specifically, our DeepMORSE integrates a deep self-expressive model with invariant expressive coefficients to discover modality-invariant expressions and learn representations that conform to the union of modality-specific subspace structures, thereby avoiding the disruption of each modality's internal distribution. The empirical validation of DeepMORSE is twofold: its shared self-expressive coefficients excel in clustering, and its learned structured representations generalize well to downstream tasks.

**Contributions.** The contributions of the paper are highlighted as follows.

1. We propose a deep **mo**dality-sha**r**ed **s**elf-**e**xpressive model (DeepMORSE) that jointly learns representations conforming to a union of modality-specific subspaces and explicitly discovers structures that are shared across modalities.

2. We demonstrate the effectiveness of our proposed DeepMORSE with extensive experiments on seven widely used image clustering benchmarks, achieving state-of-the-art performance.

3. We further validate the robustness and generality of the learned structured representations on downstream tasks including image retrieval and zero-shot classification, achieving strong performance even without task-specific optimization.

## 2 RELATED WORK

**Multi-view clustering.** Multi-view clustering (Bickel & Scheffer, 2004; Fang et al., 2023) is another clustering task that leverages multiple sources of information; however, it differs from multimodal clustering in two important aspects. First, the multiple views are provided and corresponding to the same instance, whereas multimodal clustering often requires generating auxiliary textual counterparts from images themselves. Second, the data in different views in multi-view clustering are typically homogeneous or strongly correlated, sharing a similar distribution; whereas the data in different modalities usually exhibit inherently and dramatically distinct distributions. Due to these

significant differences, existing multi-view clustering approaches—whether based on matrix factorization (Zhao et al., 2017; Wen et al., 2018; Yang et al., 2020) or subspace clustering (Cao et al., 2015; Luo et al., 2018; Li et al., 2019; Wang, 2024)—cannot be directly transferred to the multimodal setting. Although recent deep multi-view clustering methods are capable of jointly learning representations and clusters (Chen et al., 2023; Chao et al., 2024; Sun et al., 2024), these approaches generally lack explicit modeling of the intrinsic structures within the data.

**Deep clustering.** The progress of deep clustering closely follows the advances in unsupervised representation learning paradigm. The earliest deep clustering approaches are mainly based on the autoencoder (Xie et al., 2016; Guo et al., 2017; Yang et al., 2017). Then, with the flourishing success of contrastive learning (Chen et al., 2020; He et al., 2020), deep clustering approaches are designed by integrating miscellaneous information, e.g., pseudo-labeling (Van Gansbeke et al., 2020), cluster and graph-level contrastive learning (Li et al., 2021; Zhong et al., 2021; Qian et al., 2022; Qian, 2023), neighbor matching (Dang et al., 2021), prototype scattering (Huang et al., 2023), and contextually affinitive neighborhood mining (Yu et al., 2023). Among them, CoKe (Qian et al., 2022) introduces a metric-learning-style pretext task combined with an online constrained $k$-means; SeCu (Qian, 2023) effectively eliminates the direction of negative instances from the gradient for stable training. These two methods ultimately model each cluster by a centroid in the embedding space (which can be viewed as a zero-dimensional subspace). In contrast, we explicitly models clusters as subspaces. Recently, as multimodal foundation models yield powerful representations for various downstream tasks, multimodal clustering approaches have received increasing attention (Cai et al., 2023; Li et al., 2024). The essential idea of (Cai et al., 2023) and (Li et al., 2024) is to align the pseudo-labels of cross-modal data and their neighbors. Although the clustering performance improves compared with using a single modality, the distribution of the aligned representation in these methods remain unclear.

**Learning structured representations.** In the setting of supervised learning, representations that conform to a union of orthogonal subspaces are pioneered in Orthogonal Low-rank Embedding (OLE) (Lezama et al., 2018) and Maximal Coding Rate Reduction (MCR$^2$) (Yu et al., 2020). In these approaches, the learned representations for each class are guaranteed to lie on a linear subspace and the subspaces are arranged to be orthogonal. In the setting of unsupervised learning, Ding et al. (2023) presented an approach called Manifold Linearization and Clustering (MLC), which attempts to learn structured representations with modified MCR$^2$ principle where the membership information is replaced by the learned affinity. More recently, Meng et al. (2025) proposed a principled approach for deep subspace clustering (PRO-DSC), which learns structured representations by the self-expressive model with a maximal coding rate regularization and provides theoretical justifications to show that the learned representations are guaranteed to avoid the catastrophic feature collapse (Haeffele et al., 2021) and are likely to conform to the desired structure of a union of subspaces. Nevertheless these works are not designed for dealing with multi-modality data.

## 3 OUR PROPOSED APPROACH: DEEPMORSE

This section presents a deep modality-shared self-expressive approach for the learning of modal-invariant structure and the associated structured representations, and then describe the reparameterization, training procedure and text generation strategy.

### 3.1 MODALITY-SHARED DEEP SELF-EXPRESSION

We assume that the data points in each modality lie on a union of modality-specific subspaces, where the distributions may differ substantially across modalities. By modeling with modality-specific subspaces, we are able to capture the distribution diversity across modalities. Our goal is to reveal the shared underlying (partition) patterns across the data in different modalities.

**Discovering shared structures across modalities.** To learn the patterns of data, existing works typically leverage neighborhood information, either in a single-modality setting (Van Gansbeke et al., 2020; Dang et al., 2021; Niu et al., 2022; Huang et al., 2023) or in a multimodal setting (Cai et al., 2023; Li et al., 2024). While neighborhoods can faithfully reflect the local distribution of data, they are insufficient to estimate the global structure unless the data are densely sampled. As an alternative, the self-expressive model is capable of uncovering the global structure of data, because it

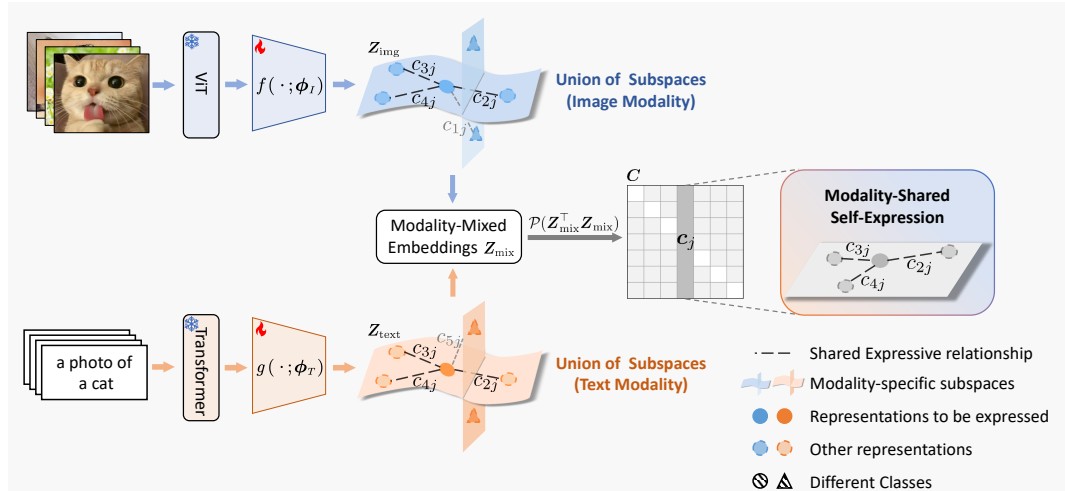

Figure 2: **The overview of DeepMORSE.** DeepMORSE learns modality-invariant structure through modality-shared self-expression, and jointly fine-tunes the representations to conform to the union of modality-specific subspace structures.

seeks the linear combination relations among proper data points rather than relying solely on local neighborhoods. Formally, the self-expressive model solves the optimization problem:

$$\min_{\boldsymbol{c}_j} \quad \|\boldsymbol{x}_j - \sum_{i \neq j} c_{ij} \boldsymbol{x}_i\|_2^2 + r(\boldsymbol{c}_j), \quad \text{for all } j \in \{1, \cdots, N\}, \tag{1}$$

where $\|\boldsymbol{x}_j - \sum_{i \neq j} c_{ij} \boldsymbol{x}_i\|_2^2$ measures the residual of expressing $\boldsymbol{x}_j$ with a linear combination of other data points, and $r(\cdot) : \mathbb{R}^N \mapsto \mathbb{R}_+$ is a regularization on the expressive coefficient $\boldsymbol{c}_j = [c_{1j}, \cdots, c_{Nj}]^\top$. After solving the problem (1), the non-zero coefficients are guaranteed to indicate which data points belong to the same subspace under certain conditions (Elhamifar & Vidal, 2013; Soltanolkotabi & Candès, 2012; Li et al., 2018).

For multimodal data, to uncover the shared underlying patterns across different modalities, it is reasonable to introduce a self-expressive model with modality-invariant coefficients $\boldsymbol{c}_j$, e.g.,

$$\min_{\boldsymbol{c}_j} \quad \|\boldsymbol{x}_j - \sum_{i \neq j} c_{ij} \boldsymbol{x}_i\|_2^2 + \|\boldsymbol{t}_j - \sum_{i \neq j} c_{ij} \boldsymbol{t}_i\|_2^2, \quad \text{for all } j \in \{1, \cdots, N\}, \tag{2}$$

where $\{\boldsymbol{t}_i\}_{i=1}^N$ are supposed to be the associated textual data. The modality-invariant constraint can be interpreted as a special regularization term applied to the coefficients for both the image and text self-expressive models, explicitly encouraging the learning of the underlying pattern shared across modalities.

**Learning modality-specific structured representations.** For the real world data, unfortunately, the distribution of the raw data in each modality might not conform to a union of subspaces. To account for more complex data and more complicated distribution structures beyond subspaces, we employ a set of transformations $f, g : \mathbb{R}^D \mapsto \mathbb{R}^d$ to learn modality-specific representations, where

$$\boldsymbol{z}_{\text{img}} = f(\boldsymbol{x}), \quad \boldsymbol{z}_{\text{text}} = g(\boldsymbol{t}). \tag{3}$$

For clarity and compactness in notions, we denote $\boldsymbol{Z}_{\text{img}} := [\boldsymbol{z}_{\text{img},1}, \cdots, \boldsymbol{z}_{\text{img},N}]$ and $\boldsymbol{Z}_{\text{text}} := [\boldsymbol{z}_{\text{text},1}, \cdots, \boldsymbol{z}_{\text{text},N}]$ as the learned representations of all images and all associated texts, respectively, and denote $\boldsymbol{C} := [\boldsymbol{c}_1, \cdots, \boldsymbol{c}_N]$ as the modality-shared self-expressive coefficient matrix. Then, for fixed $\boldsymbol{Z}_{\text{img}}$ and $\boldsymbol{Z}_{\text{text}}$, problem (2) can be equivalently reformulated as follows:

$$\min_{\boldsymbol{C}} \quad \|\boldsymbol{Z}_{\text{img}} - \boldsymbol{Z}_{\text{img}} \boldsymbol{C}\|_F^2 + \|\boldsymbol{Z}_{\text{text}} - \boldsymbol{Z}_{\text{text}} \boldsymbol{C}\|_F^2, \quad \text{s.t.} \quad \text{Diag}(\boldsymbol{C}) = \boldsymbol{0}. \tag{4}$$

Furthermore, it seems appealing to learn simultaneously the embeddings $\boldsymbol{Z}_{\text{img}}$ and $\boldsymbol{Z}_{\text{text}}$ and the coefficient matrix $\boldsymbol{C}$ by solving the problem as follows:

$$\min_{\boldsymbol{Z}_{\text{img}}, \boldsymbol{Z}_{\text{text}}, \boldsymbol{C}} \|\boldsymbol{Z}_{\text{img}} - \boldsymbol{Z}_{\text{img}}\boldsymbol{C}\|_F^2 + \|\boldsymbol{Z}_{\text{text}} - \boldsymbol{Z}_{\text{text}}\boldsymbol{C}\|_F^2, \ \ \text{s.t.} \ \ \text{Diag}(\boldsymbol{C}) = \boldsymbol{0}, \ \ \boldsymbol{Z}_{\text{img}}, \boldsymbol{Z}_{\text{text}} \in \mathbb{S}^{d-1}. \tag{5}$$

Unfortunately, such a joint optimization problem suffers from a catastrophic representation collapse issue, just as that in the unimodal deep self-expressive model pointed by (Haeffele et al., 2021). Inspired by (Meng et al., 2025), we impose on the learned modality-specific representations the maximal coding rate regularization, i.e.,

$$\rho(\boldsymbol{Z}_{\text{img}}) \coloneqq \log \det(\boldsymbol{I} + \frac{d}{N\epsilon^2}\boldsymbol{Z}_{\text{img}}\boldsymbol{Z}_{\text{img}}^\top), \quad \rho(\boldsymbol{Z}_{\text{text}}) \coloneqq \log \det(\boldsymbol{I} + \frac{d}{N\epsilon^2}\boldsymbol{Z}_{\text{text}}\boldsymbol{Z}_{\text{text}}^\top), \tag{6}$$

where $\epsilon \in \mathbb{R}_+$ is a hyperparameter. The regularization term $\rho(\cdot)$ measures the volume of space spanned by the representations (Ma et al., 2007).[1] By maximizing the volume of the learned modality-specific representations while eliminating its expressive residual, the regularized deep self-expressive model promotes the learned representations in each modality conforming to a union of the modality-specific subspaces, and the different subspaces are promoted to be orthogonal (Meng et al., 2025).

Putting these together, we have a simple but principled framework for deep multi-modal subspace clustering, called deep **mo**dality-sha**r**ed **s**elf-**e**xpressive model (DeepMORSE):

$$\min_{\boldsymbol{Z}_{\text{img}}, \boldsymbol{Z}_{\text{text}}, \boldsymbol{C}} \gamma \left( \|\boldsymbol{Z}_{\text{img}} - \boldsymbol{Z}_{\text{img}}\boldsymbol{C}\|_F^2 + \|\boldsymbol{Z}_{\text{text}} - \boldsymbol{Z}_{\text{text}}\boldsymbol{C}\|_F^2 \right) - \rho(\boldsymbol{Z}_{\text{img}}) - \rho(\boldsymbol{Z}_{\text{text}}),$$
$$\text{s.t.} \quad \text{Diag}(\boldsymbol{C}) = \boldsymbol{0}, \ \ \boldsymbol{Z}_{\text{img}}, \boldsymbol{Z}_{\text{text}} \in \mathbb{S}^{d-1} \tag{7}$$

where $\gamma \in \mathbb{R}_+$ is a balancing hyperparameter. By optimizing DeepMORSE, modality-shared structures are captured by the self-expressive matrix $\boldsymbol{C}$; simultaneously, the multimodal representations $\{\boldsymbol{Z}_{\text{img}}, \boldsymbol{Z}_{\text{text}}\}$ are fine-tuned to conform to the union of modality-specific subspace structures.

### 3.2 IMPLEMENTATIONS

**Reparameterization and training.** To learn scalable and generalizable representations, we use MLPs to reparameterize the learnable transformations, i.e., $f(\,\cdot\,; \boldsymbol{\phi}_I)$ and $g(\,\cdot\,; \boldsymbol{\phi}_T)$, where $\boldsymbol{\phi}_I$ and $\boldsymbol{\phi}_T$ denote the learnable parameters. Accordingly, the image and text representations are obtained by:

$$\boldsymbol{z}_{\text{img}} = f(\boldsymbol{x}; \boldsymbol{\phi}_I)/\|f(\boldsymbol{x}; \boldsymbol{\phi}_I)\|_2, \quad \boldsymbol{z}_{\text{text}} = g(\boldsymbol{t}; \boldsymbol{\phi}_T)/\|g(\boldsymbol{t}; \boldsymbol{\phi}_T)\|_2, \tag{8}$$

where the normalization is to enforce the constraint on representations in (5).

In addition, since the number of expressive coefficients $\{c_{ij}\}_{i,j=1}^N$ grows quadratically with $N$, we also follow (Zhang et al., 2021; Meng et al., 2025) to reparameterize the coefficients with a properly designed two-branch network. Specifically, we first compute a modality-mixed representation $\boldsymbol{z}_{\text{mix}} \coloneqq \frac{1}{2}(\boldsymbol{z}_{\text{img}} + \boldsymbol{z}_{\text{text}})$, from which the expressive coefficients are obtained as:

$$\boldsymbol{C} = \mathcal{P}(\boldsymbol{Z}_{\text{mix}}^\top \boldsymbol{Z}_{\text{mix}}), \tag{9}$$

where $\mathcal{P}(\cdot)$ denotes the Sinkhorn-Knopp projection (Cuturi, 2013), commonly used to normalize membership matrices in clustering (Caron et al., 2020; Ding et al., 2022; 2023).[2]

For optimization, we jointly train all parameters $\{\boldsymbol{\phi}_I, \boldsymbol{\phi}_T\}$ via stochastic gradient descent (SGD) with the following loss function:

$$\mathcal{L} \coloneqq \gamma \left( \|\boldsymbol{Z}_{\text{img}} - \boldsymbol{Z}_{\text{img}}\boldsymbol{C}\|_F^2 + \|\boldsymbol{Z}_{\text{text}} - \boldsymbol{Z}_{\text{text}}\boldsymbol{C}\|_F^2 \right) - \rho(\boldsymbol{Z}_{\text{img}}) - \rho(\boldsymbol{Z}_{\text{text}}). \tag{10}$$

After training, clustering results are obtained through spectral clustering (Shi & Malik, 2000) on the affinity matrix induced by $|\boldsymbol{Z}_{\text{img}}^\top \boldsymbol{Z}_{\text{img}}|$. Moreover, the structured multimodal representations $\boldsymbol{Z}_{\text{img}}$

---

[1]The volume of space is estimated by counting the number of $\epsilon$-balls which can be packed (Ma et al., 2007).

[2]After applying the Sinkhorn-Knopp projection, we eliminate the diagonal elements of $\boldsymbol{C}$ to satisfy the constraint.

and $\boldsymbol{Z}_{\text{text}}$ learned by DeepMORSE can be directly applied to downstream tasks such as zero-shot classification and image retrieval.

**Text Counterpart Generation.** In practical image clustering scenarios, off-the-shelf image-text pairs are seldom available. Therefore, we generate a textual counterpart for each image in order to leverage textual information. A desirable textual counterpart should accurately capture the semantic content of the image while maintaining a simple and clear geometric structure. Formally, denote $\boldsymbol{x} \in \mathbb{R}^D$ as an image embedding, $\mathcal{D} = \{\boldsymbol{\pi}_1, \cdots, \boldsymbol{\pi}_M\}$ as a dictionary of text embeddings.[3] We generate a textual counterpart $\boldsymbol{t} \in \mathbb{R}^D$ by solving the following cross-modal sparse coding problem:

$$\min_{\boldsymbol{t}} \quad \|\boldsymbol{x} - \boldsymbol{t}\|_2^2, \qquad \text{s.t.} \qquad \boldsymbol{t} = \sum_{i=1}^M \theta_i \boldsymbol{\pi}_i, \ \|\boldsymbol{\theta}\|_0 \leq s, \tag{11}$$

where $\|\boldsymbol{\theta}\|_0$ denotes the number of non-zero coefficients. By minimizing the Euclidean distance between $\boldsymbol{x}$ and $\boldsymbol{t}$, the describable preciseness of the textual counterpart is enhanced. The sparsity constraint enforces that only the top-$s$ relevant semantics are preserved, so that the generated textual counterpart $\boldsymbol{t}$ lies in the *subspace* spanned by a small set of semantic embeddings of $\mathcal{D}$. By solving problem (11) for all images $\mathcal{X} := \{\boldsymbol{x}_1, \cdots, \boldsymbol{x}_N\}$, we obtain the respective optimal textual counterparts $\{\boldsymbol{t}_i\}_{i=1}^N$, which accurately reflect the content of each image while maintaining a union-of-subspaces structure. We present the detailed algorithm of our text counterpart generation in the Appendix A.

The overall framework of DeepMORSE is shown in Figure 2, and the training and testing procedures are summarized in Algorithm 1.

---

**Algorithm 1** Deep Modality-Shared Self-Expressive Model (DeepMORSE)

---

**Input:** Image embeddings $\mathcal{X}$, dictionary of text embeddings $\mathcal{D}$, hyperparameters $s, \gamma, \epsilon$, number of epochs $T$, learning rate $\eta$
**Text generation:** Generate text counterparts by solving problem (11) through matching pursuit
**Initialization:** Randomly initialize network parameters $\boldsymbol{\phi}_I, \boldsymbol{\phi}_T$
1: **for** $t = 1, \ldots, T$ **do**
2:     *# Forward propagation*
3:     Compute representations $\boldsymbol{z}_{\text{img}}$ and $\boldsymbol{z}_{\text{text}}$ by (8)
4:     Compute modality-shared self-expressive matrix $\boldsymbol{C}$ by (9)
5:     *# Backward propagation*
6:     Compute loss $\mathcal{L}$ by (10)
7:     Compute gradient $\nabla_{\boldsymbol{\phi}_I} := \frac{\partial \mathcal{L}}{\partial \boldsymbol{\phi}_I}$, $\nabla_{\boldsymbol{\phi}_T} := \frac{\partial \mathcal{L}}{\partial \boldsymbol{\phi}_T}$
8:     Set $\boldsymbol{\phi}_I \leftarrow \boldsymbol{\phi}_I - \eta \cdot \nabla_{\boldsymbol{\phi}_I}$, $\boldsymbol{\phi}_T \leftarrow \boldsymbol{\phi}_T - \eta \cdot \nabla_{\boldsymbol{\phi}_T}$
9: **end for**
**Test:** Apply spectral clustering on $|\boldsymbol{Z}_{\text{img}}^\top \boldsymbol{Z}_{\text{img}}|$.

---

## 4 EXPERIMENTS

In this section, we will first demonstrate the experimental setup of DeepMORSE (Section 4.1), then validate its effectiveness by presenting its clustering performance (Section 4.2) and showing its transferability of learned representations on downstream tasks (Section 4.3).

### 4.1 EXPERIMENTAL SETUP

Unless otherwise specified, we fix the hyperparameters to $s = 5$, $\gamma = 150$, and $\epsilon = 0.1$ on all datasets. The reparameterized transformations $f(\,\cdot\,; \phi_I)$ and $g(\,\cdot\,; \phi_T)$ are implemented as two-layer MLPs with hidden dimension 512 and output dimension 128. DeepMORSE is trained with a learning rate of $\eta = 10^{-4}$ and a batch size of $n_b = 1024$. We evaluate clustering performance using accuracy (ACC) and normalized mutual information (NMI), and report the average results over five

---

[3] Following (Li et al., 2024), the dictionary is built by a filtered WordNet. Refer to Appendix A for details.

Table 1: **Image clustering performance comparing to the baselines.** The best results are in **bold** and the second best results are underlined. The method marked with † is based on our reproduction, due to the distinction of architectures.

| | Backbone | Text Modality | CIFAR-10 ACC | CIFAR-10 NMI | CIFAR-20 ACC | CIFAR-20 NMI | Dogs-15 ACC | Dogs-15 NMI | DTD-47 ACC | DTD-47 NMI | UCF-101 ACC | UCF-101 NMI |
|---|---|---|---|---|---|---|---|---|---|---|---|---|
| GCC (Zhong et al., 2021) | ResNet-18 | ✗ | 85.6 | 76.4 | 47.2 | 47.2 | 52.6 | 49.0 | - | - | - | - |
| NNM (Dang et al., 2021) | ResNet-18 | ✗ | 83.7 | 73.7 | 45.9 | 48.0 | 58.6 | 60.4 | - | - | - | - |
| SCAN (Van Gansbeke et al., 2020) | ResNet-18 | ✗ | 88.3 | 79.7 | 50.7 | 48.6 | 59.3 | 61.2 | 46.4 | 59.4 | 61.1 | 79.7 |
| CoKe (Qian et al., 2022) | ResNet-18 | ✗ | 85.7 | 76.6 | 49.7 | 49.1 | - | - | - | - | - | - |
| SeCu (Qian, 2023) | ResNet-18 | ✗ | **93.0** | **86.1** | 55.2 | 55.1 | - | - | - | - | - | - |
| PRO-DSC† (Meng et al., 2025) | ViT-B/32 | ✗ | 86.7 | 79.1 | 58.8 | 62.1 | 32.3 | 25.8 | 48.1 | 57.0 | 68.0 | 83.6 |
| CLIP ($k$-means) (Radford et al., 2021) | ViT-B/32 | ✗ | 74.2 | 70.3 | 45.5 | 49.9 | 38.1 | 39.8 | 42.6 | 57.3 | 58.2 | 79.5 |
| SIC (Cai et al., 2023) | ViT-B/32 | ✓ | 92.6 | 84.7 | 58.3 | 59.3 | 69.7 | 69.0 | 45.9 | 59.6 | 61.9 | 81.0 |
| TAC (Li et al., 2024) | ViT-B/32 | ✓ | 91.9 | 83.3 | 60.7 | 61.1 | 83.0 | 80.6 | 50.1 | 62.1 | 68.7 | 82.3 |
| Our DeepMORSE | ViT-B/32 | ✓ | 92.3 | 84.1 | **63.3** | **63.1** | **88.7** | **84.4** | **55.0** | **64.9** | **73.2** | **86.1** |

random seeds.[4] For downstream tasks, we adopt mean average precision (mAP) for image retrieval and accuracy (ACC) for zero-shot classification. Please refer to Appendix B for more experimental details.

## 4.2 MAIN RESULTS

**Clustering performance.** We conduct experiments on seven image clustering benchmarks, including CIFAR-10 (Krizhevsky et al., 2009), CIFAR-20 (Krizhevsky et al., 2009), STL-10 (Coates et al., 2011), ImageNet-10 (Chang et al., 2017), ImageNet-Dogs (Chang et al., 2017), DTD-47 (Cimpoi et al., 2014), and UCF-101 (Soomro et al., 2012). Since clustering performance on STL-10 and ImageNet-10 has already saturated ($> 98\%$), we defer the comparison results on these datasets to Appendix C. We compare DeepMORSE with unimodal deep clustering baselines (e.g., GCC (Zhong et al., 2021), NNM (Dang et al., 2021), SCAN (Van Gansbeke et al., 2020), CoKe (Qian et al., 2022), SeCu (Qian, 2023), PRO-DSC (Meng et al., 2025)) as well as multimodal methods (e.g., SIC (Cai et al., 2023), TAC (Li et al., 2024)). For "CLIP ($k$-means)", we compute image embeddings with the pretrained CLIP image encoder and then directly run $k$-means algorithm on these embeddings. For each method, we also report its backbone architecture and whether it leverages textual information.

As shown in Table 1, DeepMORSE achieves state-of-the-art clustering performance, particularly on more challenging datasets. Specifically, it improves clustering accuracy by $4.5\%$, $4.9\%$, and $5.7\%$ on UCF-101, DTD-47, and ImageNet-Dogs, respectively. In addition, since PRO-DSC is a deep self-expressive model for unimodal image clustering, the improvements of DeepMORSE over PRO-DSC provide clear evidence that textual information can be effectively leveraged to enhance clustering.

**Ablation studies.** To evaluate the contribution of each component in DeepMORSE, we conduct ablation studies and report the results in Table 2. First, we evaluate unimodal variants by removing the loss from the other modality.[5] As shown in lines 1 and 2, clustering performance degrades sharply when relying on a single modality. Removing the representation regularization terms further causes severe degeneration (line 3), consistent with the collapse problem reported in (Haeffele et al., 2021; Meng et al., 2025). Finally, when replacing our text generation method with neighborhood-based retrieval (Li et al., 2024), the results still surpass those of (Li et al., 2024), highlighting the robustness of our modality-shared self-expression (line 4). [6]

**Visualization of self-expressive matrix.** To visualize the shared structure captured by Deep-MORSE, we present the learned self-expressive matrices $C$ on CIFAR-10 (first row) and ImageNet-Dogs (second row) at epochs 3, 10, and 50. For comparison, we also illustrate the Gram matrices of the input CLIP image embeddings $X_{\text{img}}$ and their textual counterparts $X_{\text{text}}$. In addition, the clustering accuracy is also reported above the corresponding Gram and self-expressive matrices.

---

[4]Standard deviations are provided in Appendix C.

[5]Unimodal self-expressive matrices are defined as $C_{\text{img}} := \mathcal{P}(Z_{\text{img}}^\top Z_{\text{img}})$ and $C_{\text{text}} := \mathcal{P}(Z_{\text{text}}^\top Z_{\text{text}})$, with $\gamma$ selected from $\{0.5, 1, 5, 10, 50, 100, 150, 300, 500\}$.

[6]Besides, we also compare against alternative ways of combining self-expressive matrices from both modalities (e.g., summation or element-wise maximum), which do not enforce modality-shared coefficients. The results are provided in Appendix C.

Table 2: **Ablation studies.** Rows 1 and 2: unimodal results; Row 3: without representation regularization; Row 4: neighborhood-based textual retrieval.

| Text generation | | Deep self-expression | | Regularization | | Clustering performance (ACC%) | | | | |
| --- | --- | --- | --- | --- | --- | --- | --- | --- | --- | --- |
| Neighbor | Sparse coding | $\mathcal{L}_{\text{img}}^{\text{expr}}$ | $\mathcal{L}_{\text{text}}^{\text{expr}}$ | $\rho(\boldsymbol{Z}_{\text{img}})$ | $\rho(\boldsymbol{Z}_{\text{text}})$ | CIFAR-10 | CIFAR-20 | ImgNet-Dogs | DTD-47 | UCF-101 |
| | | ✓ | | ✓ | | 86.7 | 58.8 | 32.3 | 48.1 | 68.0 |
| | | ✓ | ✓ | | ✓ | 71.2 | 34.5 | 72.9 | 40.2 | 60.7 |
| | | ✓ | ✓ | | | 33.0 | 9.4 | 17.2 | 27.0 | 47.8 |
| ✓ | | ✓ | ✓ | ✓ | ✓ | 92.0 | 62.4 | 88.5 | 54.3 | 72.2 |
| | ✓ | ✓ | ✓ | ✓ | ✓ | **92.3** | **63.3** | **88.7** | **55.0** | **73.2** |

(a) $|\boldsymbol{X}_{\text{img}}^{\top}\boldsymbol{X}_{\text{img}}|$ (86.7%, 32.3%)  (b) $|\boldsymbol{X}_{\text{text}}^{\top}\boldsymbol{X}_{\text{text}}|$ (71.2%, 72.9%)  (c) $\boldsymbol{C}^{(3)}$ (91.4%, 72.9%)  (d) $\boldsymbol{C}^{(10)}$ (92.4%, 79.3%)  (e) $\boldsymbol{C}^{(50)}$ (92.5%, 88.0%)

Figure 3: **Comparison between Gram matrices of image/text embeddings and the modality-shared self-expressive matrices learned by DeepMORSE.** First row: CIFAR-10; second row: ImageNet-Dogs. Numbers above each matrix indicate clustering accuracy.

As shown in Figure 3, the self-expressive matrices learned by DeepMORSE progressively evolve toward block-diagonal structures, indicating that intra-class similarities are progressively amplified while inter-class similarities are suppressed as training proceeds.

**Sensitivity of hyperparameters.** To discover the influence of hyperparameters on clustering performance, we conduct a set of experiments with varying $s$ and $\gamma$ on all image clustering benchmarks used in this work and illustrate the results in Figure 4. Additionally, to make the robustness assessment more convincing and statistically sound, we also repeat each experiment five trials with different random seeds and plot the mean performance together with the standard deviation (shown as shaded areas around the curves). For comparison, we also illustrate the best clustering result obtained without using the text modality. As can be observed, on most datasets, DeepMORSE outperforms the unimodal baselines across different settings of $s$ and $\gamma$, demonstrating its robustness to hyperparameter choices. Notably, the performance of DeepMORSE exhibits the least robustness on ImageNet-Dogs, which may be attributed to the significant performance gap between its two modalities (as shown in Table 2: 32.3%/72.9%).

**Visualization via PCA.** To visualize the structure of representations learned by TAC (Li et al., 2024) and DeepMORSE, we illustrate the dimension reduction results via PCA of the learned representations for each modality on all benchmark datasets. Note that PCA preserves the global structure of the representations as it reduces dimension by learning a *linear* projection of the input data. As shown in Figure 5, the representations learned by DeepMORSE (rows 3 and 4) form approximately a union of orthogonal low-dimensional subspaces, where representations from each class are concentrated within a distinct subspace. For TAC (rows 1 and 2), the projected embeddings exhibit a variety of structures that cannot be easily approximated by a union of centroids, subspaces, or manifolds.

### 4.3 EVALUATIONS ON REPRESENTATIONS

Compared with existing deep multimodal clustering approaches, an intriguing advantage of DeepMORSE lies in its learned multimodal representations $\{\boldsymbol{Z}_{\text{img}}, \boldsymbol{Z}_{\text{text}}\}$, which conform to the discovered shared subspace structure. To validate the effectiveness of these representations, we transfer

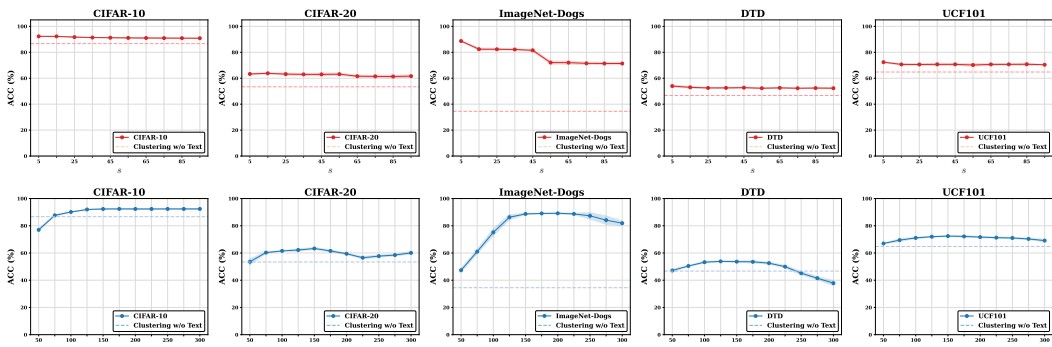

Figure 4: **Sensitivity of hyper-parameters $s$ (top) and $\gamma$ (bottom).** Solid lines: clustering accuracy (ACC: %) with varying $s$ and $\gamma$; dashed lines: the best unimodal counterpart without textual data.

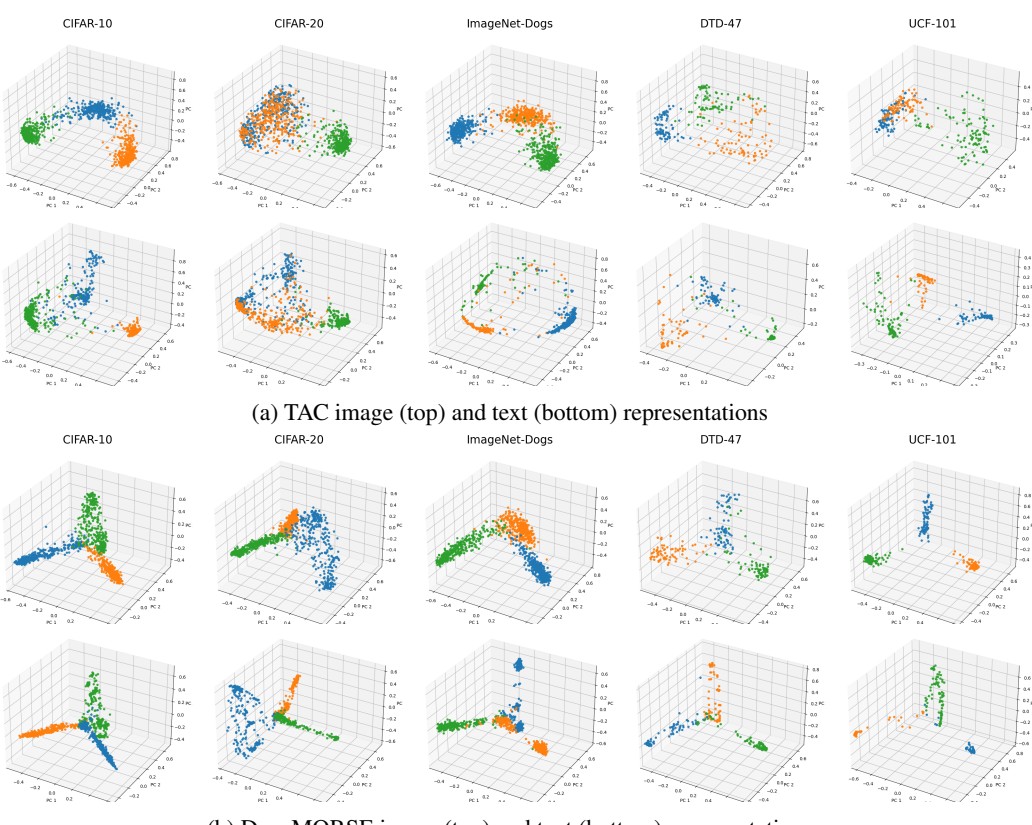

(a) TAC image (top) and text (bottom) representations

(b) DeepMORSE image (top) and text (bottom) representations

Figure 5: **Visualization of image and text representations via PCA** which are learned by TAC (rows 1 and 2) and DeepMORSE (rows 3 and 4).

them to two downstream tasks and compare their performance with state-of-the-art approaches in each field. It is noteworthy that DeepMORSE does not require additional loss functions or task-specific postprocessing for these downstream evaluations, and the hyperparameter configuration is kept the same for these two tasks.

We evaluate on multiple image classification and retrieval benchmarks, including Flowers102 (Nilsback & Zisserman, 2008), DTD (Cimpoi et al., 2014), OxfordPets (Parkhi et al., 2012), StanfordCars (Krause et al., 2013), UCF101 (Soomro et al., 2012), Caltech101 (Fei-Fei et al., 2004), Food101 (Fei-Fei et al., 2004), SUN397 (Xiao et al., 2010), FGVCAircraft (Maji et al., 2013), and EuroSAT (Helber et al., 2019). For StanfordCars and SUN397, which contain more than 128 cate-

gories, we increase the output dimensions to 256 and 512, respectively. Accordingly, the balancing hyperparameter $\gamma$ is also enlarged for these two datasets.

**Image retrieval.** Image retrieval is a unimodal downstream task that aims to retrieve images belonging to the same category. Here, we directly perform the retrieval based on the cosine similarity of the image representations learned by DeepMORSE.

As shown in Table 3, DeepMORSE achieves state-of-the-art performance on image retrieval without any task-specific design. Prior work (Mistretta et al., 2025) has shown that intra-modal similarity scores from CLIP models do not always correspond faithfully to semantic similarities among images. In contrast, the representations learned by DeepMORSE mitigate this issue by aligning with structures shared across modalities.

Table 3: **Image retrieval performance (mAP: %) comparing to the state-of-the-art.** The comparative results are directly cited. The image encoder of all methods are based on ViT-B/32.

| Method | Flower | DTD | Pets | Cars | UCF | Caltech | Food | SUN | Aircraft | EuroSAT |
|---|---|---|---|---|---|---|---|---|---|---|
| CLIP ViT-B/32 (Radford et al., 2021) | 62.0 | 28.1 | 30.5 | 24.6 | 47.1 | 77.1 | 32.3 | 34.3 | 14.5 | 47.9 |
| OTI (Mistretta et al., 2025) | 62.6 | 31.9 | 37.5 | 28.0 | 48.6 | 79.9 | 34.7 | 36.3 | 14.4 | 47.2 |
| Our DeepMORSE | **65.2** | **32.0** | **57.7** | **28.9** | **53.3** | **81.1** | **57.1** | **42.6** | **15.6** | **53.4** |

**Zero-shot classification.** Zero-shot classification is a multimodal downstream task that assigns each image to a class based on the similarity between its representation and the text representations of the given labels. Since the categories of the datasets are known, we construct the dictionary $\mathcal{D} = \{\boldsymbol{\pi}_1, \cdots, \boldsymbol{\pi}_C\}$ for cross-modal sparse coding using embeddings of the prompts "A photo of [class]" extracted by a pretrained CLIP text encoder, where $C$ is the number of classes. After training DeepMORSE, each image $\boldsymbol{x}$ is classified according to the similarity between its learned representation and the class representations:

$$\underset{j \in \{1, \cdots, C\}}{\arg\max} \quad \frac{f(\boldsymbol{x})^\top g(\boldsymbol{\pi}_j)}{\|f(\boldsymbol{x})\|_2 \|g(\boldsymbol{\pi}_j)\|_2}. \tag{12}$$

The results are reported in Table 4, where the baseline results are directly cited from (Li et al., 2025). As shown, DeepMORSE achieves performance comparable to baselines and even surpasses the state-of-the-art on datasets Flowers102, OxfordPets, and StanfordCars—without any task-specific design for zero-shot classification.

Table 4: **Zero-shot classification performance (ACC: %) comparing to the state-of-the-art.** The comparative results are directly cited. The image encoder of all methods are based on ViT-B/16.

| Method | Flower | DTD | Pets | Cars | UCF | Caltech | Food | SUN | Aircraft | EuroSAT |
|---|---|---|---|---|---|---|---|---|---|---|
| CLIP ViT-B/16 (Radford et al., 2021) | 64.4 | 44.3 | 88.3 | 65.5 | 65.1 | 93.4 | 83.7 | 62.6 | 23.7 | 42.0 |
| Ensemble (Zhang et al., 2022) | 67.0 | 45.0 | 86.9 | 66.1 | 65.2 | 93.6 | 82.9 | 65.6 | 23.2 | 50.4 |
| TPT (Shu et al., 2022) | 69.0 | 47.8 | 87.8 | 66.9 | 68.0 | 94.2 | 84.7 | 65.5 | 24.8 | 42.4 |
| DiffTPT (Feng et al., 2023) | 70.1 | 47.0 | 88.2 | 67.0 | 68.2 | 92.5 | **87.2** | 65.7 | 25.6 | 43.1 |
| DMN (Zhang et al., 2024) | 75.3 | 54.9 | 91.2 | 67.0 | 72.0 | 93.6 | 84.1 | 69.1 | 28.3 | 56.2 |
| TDA (Karmanov et al., 2024) | 71.4 | 47.4 | 88.6 | 67.3 | 70.7 | 94.2 | 86.1 | 67.6 | 23.9 | **58.0** |
| ZLaP (Kalantidis et al., 2024) | 73.5 | 48.6 | 87.1 | 65.6 | 71.5 | 93.1 | 86.9 | 67.4 | 25.4 | 55.6 |
| ECALP (Li et al., 2025) | 76.0 | **56.3** | 92.3 | 68.2 | **75.4** | **94.4** | 85.7 | **70.5** | **29.5** | 56.5 |
| Our DeepMORSE | **76.9** | 54.8 | **93.0** | **68.7** | 74.1 | 91.9 | 86.2 | 70.2 | 27.1 | 44.4 |

## 5 CONCLUSION

We have presented a simple but principled approach for deep image clustering assisted with textual information, called DeepMORSE, which jointly learns representations that conform to a union of modality-specific subspaces and discovers shared expressions across modalities. Its effectiveness has been demonstrated through strong performance on clustering and competitive results on downstream tasks.

**Limitations.** The theoretical underpinnings of modality-shared self-expression remain largely unexplored, leaving the working mechanism insufficiently understood.

## ETHICS STATEMENT

In this work, we aim to extend traditional subspace clustering algorithms by leveraging deep learning techniques to enhance their representation learning capabilities. Our research does not involve any human subjects, and we have carefully ensured that it poses no potential risks or harms. Additionally, there are no conflicts of interest, sponsorship concerns, or issues related to discrimination, bias, or fairness associated with this study. We have taken steps to address privacy and security concerns, and all data used comply with legal and ethical standards. Our work fully adheres to research integrity principles, and no ethical concerns have arisen during the course of this study.

## REPRODUCIBILITY STATEMENT

To ensure the reproducibility of our work, we have released the source code. All datasets used in our experiments are publicly available, and we have provided a comprehensive description of the data processing steps in Appendix B. Additionally, detailed experimental settings and configurations are outlined in Appendix B to facilitate the reproduction of our results.

## THE USE OF LARGE LANGUAGE MODELS (LLMS)

In accordance with the ICLR 2026 policy, we confirm that no large language models (LLMs) were used at any stage of this work. All aspects of the research process—including problem formulation, methodology design, experimentation, analysis, and manuscript preparation—were carried out solely by the authors. The results, discussions, and conclusions presented in this paper are entirely based on the authors' own contributions, without reliance on any generative AI tools. The authors take full responsibility for the content under their names.

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

# SUPPLEMENTARY MATERIAL FOR "LEARNING DEEP MODALITY-SHARED SELF-EXPRESSIVENESS FOR IMAGE CLUSTERING WITH TEXTUAL INFORMATION"

## A  DETAILS FOR TEXT GENERATION

**Construction of dictionary** $\mathcal{D}$**.**  Since WordNet contains 146,347 words with heterogeneous semantic meanings, we follow TAC (Li et al., 2024) to filter the vocabulary before text generation. Specifically, after extracting image and text embeddings using a pretrained vision-language model, we apply $k$-means clustering to the image embeddings to obtain $k$ cluster centers. For each cluster center, the top-$\gamma$ most similar word embeddings are then selected to construct the dictionary for cross-modal sparse coding. Following TAC (Li et al., 2024), we fix $k = N/300$ and $\gamma = 5$ for all datasets.

**Detailed algorithm for text generation.**  We adopt matching pursuit (Mallat & Zhang, 1993) to solve the cross-modal sparse coding problem 11. Unlike the typical setting of sparse coding, where the query and the dictionary belong to the same modality, here the query comes from the image modality while the dictionary is constructed from the text modality. The detailed procedure is summarized in Algorithm A.1.

---

**Algorithm A.1** Cross-modal Matching Pursuit for Text Counterparts Generation

---

**Input:** Image feature $\boldsymbol{x} \in \mathbb{R}^D$, sparsity hyperparameter $s$, text features $\boldsymbol{\Pi} = [\boldsymbol{\pi}_1, \cdots, \boldsymbol{\pi}_M]$
**Initialization:** Expressive coefficients $\boldsymbol{\theta}_0 \leftarrow \mathbf{0}_M$
1: **for** $k = 1, \cdots, s$ **do**
2:     $\boldsymbol{r}_k \leftarrow \boldsymbol{\Pi}\boldsymbol{\theta}_k - \boldsymbol{x}$
3:     $i_k \leftarrow \arg\max_i |\boldsymbol{\pi}_i^\top \boldsymbol{r}_k|$
4:     $t_k \leftarrow -\frac{\langle \boldsymbol{\pi}_{i_k}, \boldsymbol{r}_k \rangle}{\|\boldsymbol{\pi}_{i_k}\|_2}$
5:     $\boldsymbol{\theta}_{k+1} \leftarrow \boldsymbol{\theta}_k + t_k \boldsymbol{e}_{i_k}$
6: **end for**
**Output:** Text counterpart $\boldsymbol{t} \leftarrow \boldsymbol{\Pi}\boldsymbol{\theta}_K$.

---

## B  EXPERIMENTAL DETAILS

We will demonstrate the dataset preprocessing, experimental details, and hyperparameter configuration for each task.

**Image clustering.**  For a fair comparison, we follow the preprocessing steps of TAC (Li et al., 2024). Specifically, images are first resized to 224 pixels on the shorter side (256 for ImageNet-10 and ImageNet-Dogs), then center-cropped to $224 \times 224$, and finally fed into the CLIP ViT-B/32 pretrained image encoder to obtain image embeddings. The construction of text embedding dictionary $\mathcal{D}$ also follows TAC (Li et al., 2024), as detailed in Section A. With the query image embeddings and the text embedding dictionary $\mathcal{D}$, textual counterparts are generated via cross-modal sparse coding, as summarized in Algorithm A.1.

We summarize the hyperparameters used to train DeepMORSE in Table B.1. The model is trained with stochastic gradient descent (SGD) using a learning rate of $\eta = 1 \times 10^{-4}$, weight decay $wd = 1 \times 10^{-4}$, and batch size $n_b = 1024$. The hidden dimension $d_{\text{hidden}} = 512$ and output dimension $d = 128$ of $f(\,\cdot\,;\phi_I)$ and $g(\,\cdot\,;\phi_T)$ are fixed across all datasets. Similarly, the sparsity level of sparse coding is set to $s = 5$, and the loss hyperparameters are fixed to $\gamma = 150$ and $\epsilon^2 = 0.1$ for all datasets.

**Image retrieval.**  For a fair comparison, we directly use the image embeddings provided by OTI (Mistretta et al., 2025), which were extracted using the pretrained CLIP ViT-B/32 model. To generate the textual counterpart of each image embedding, we first construct the text embedding dictionary as described in Section A, and then solve the cross-modal sparse coding problem with

Table B.1: **Hyperparameters configuration for training DeepMORSE for image clustering.**

| | $\eta$ | $wd$ | $d_{\text{hidden}}$ | $d$ | $T$ | $n_b$ | $\gamma$ | $\epsilon^2$ | $s$ |
|---|---|---|---|---|---|---|---|---|---|
| CIFAR-10 | $1 \times 10^{-4}$ | $1 \times 10^{-4}$ | 512 | 128 | 50 | 1024 | 150 | 0.1 | 5 |
| CIFAR-20 | $1 \times 10^{-4}$ | $1 \times 10^{-4}$ | 512 | 128 | 50 | 1024 | 150 | 0.1 | 5 |
| STL-10 | $1 \times 10^{-4}$ | $1 \times 10^{-4}$ | 512 | 128 | 50 | 1024 | 150 | 0.1 | 5 |
| ImageNet-10 | $1 \times 10^{-4}$ | $1 \times 10^{-4}$ | 512 | 128 | 50 | 1024 | 150 | 0.1 | 5 |
| ImageNet-Dogs | $1 \times 10^{-4}$ | $1 \times 10^{-4}$ | 512 | 128 | 50 | 1024 | 150 | 0.1 | 5 |
| DTD-47 | $1 \times 10^{-4}$ | $1 \times 10^{-4}$ | 512 | 128 | 100 | 1024 | 150 | 0.1 | 5 |
| UCF-101 | $1 \times 10^{-4}$ | $1 \times 10^{-4}$ | 512 | 128 | 100 | 1024 | 150 | 0.1 | 5 |

Algorithm A.1. After training DeepMORSE, image retrieval is performed based on the cosine similarity among the learned image representations.

We summarize the hyperparameters for training DeepMORSE on the image retrieval task in Table B.2, where most hyperparameters remain unchanged, except for extended epochs and larger $d, \gamma$ for datasets with greater than 128 classes. Specifically, since the benchmark datasets are relatively small in size (for example, OxfordPets, Flowers102, FGVCAircraft, DTD, and Caltech101 each contain fewer than 5,000 training samples), we extend the training epochs to ensure sufficient iterations. For StanfordCars and SUN397, which include more than 128 categories, we increase the hidden and output dimensions of the model and adjust $\gamma$ accordingly to balance the different loss terms.

Table B.2: **Hyperparameters configuration for training DeepMORSE for image retrieval and zero-shot classification.**

| | $\eta$ | $wd$ | $d_{\text{hidden}}$ | $d$ | $T$ | $n_b$ | $\gamma$ | $\epsilon^2$ | $s$ |
|---|---|---|---|---|---|---|---|---|---|
| Flower | $1 \times 10^{-4}$ | $1 \times 10^{-4}$ | 512 | 128 | 200 | 1024 | 150 | 0.1 | 5 |
| DTD | $1 \times 10^{-4}$ | $1 \times 10^{-4}$ | 512 | 128 | 200 | 1024 | 150 | 0.1 | 5 |
| Pets | $1 \times 10^{-4}$ | $1 \times 10^{-4}$ | 512 | 128 | 200 | 1024 | 150 | 0.1 | 5 |
| UCF | $1 \times 10^{-4}$ | $1 \times 10^{-4}$ | 512 | 128 | 200 | 1024 | 150 | 0.1 | 5 |
| Caltech | $1 \times 10^{-4}$ | $1 \times 10^{-4}$ | 512 | 128 | 200 | 1024 | 150 | 0.1 | 5 |
| Food | $1 \times 10^{-4}$ | $1 \times 10^{-4}$ | 512 | 128 | 200 | 1024 | 150 | 0.1 | 5 |
| Aircraft | $1 \times 10^{-4}$ | $1 \times 10^{-4}$ | 512 | 128 | 200 | 1024 | 150 | 0.1 | 5 |
| EuroSAT | $1 \times 10^{-4}$ | $1 \times 10^{-4}$ | 512 | 128 | 200 | 1024 | 150 | 0.1 | 5 |
| Cars | $1 \times 10^{-4}$ | $1 \times 10^{-4}$ | 512 | 256 | 1000 | 1024 | 200 | 0.1 | 5 |
| SUN | $1 \times 10^{-4}$ | $1 \times 10^{-4}$ | 2048 | 512 | 2000 | 1024 | 350 | 0.1 | 5 |

Here, we present a simple and principled rule for tuning hyperparameters:

1. Adjusting the output dimension $d$. DeepMORSE encourages representations from different classes to occupy orthogonal subspaces. Thus, the output dimension must be greater than the number of classes to accommodate these subspaces.

2. Adjusting the balancing hyperparameter $\gamma$. As justified in (Meng et al., 2025), the upper bound of $\gamma$ scales linearly with $\alpha = d/(N\epsilon^2)$, where $d$ is the output dimension, $N$ is the batch-size, and $\epsilon$ is the coding precision. Since that DeepMORSE uses the same $N$ and $\epsilon$ for all experiments, $\gamma$ scales linearly with the output dimension $d$. Empirically, our settings for $\gamma$ and $d$ approximately follow $\gamma \approx 0.5 \times d + 80$, which provides a simple rule that can be used on other datasets without manual search.

**Zero-shot classification.** For a fair comparison with existing approaches, we directly use the image embeddings provided by ECALP (Li et al., 2025), which are extracted using the pretrained CLIP ViT-B/16 model. Since the dataset categories are known in zero-shot classification, we construct the text embedding dictionary $\mathcal{D} = \{\pi_1, \cdots, \pi_C\}$ by extracting text embeddings of the prompts "A photo of class" with a pretrained CLIP text encoder, where $C$ denotes the number of classes. The textual counterpart of each image embedding is then generated using Algorithm A.1. For training

DeepMORSE, the hyperparameter configuration is kept exactly the same as in the image retrieval task.

**Reproducing PRO-DSC.** The original PRO-DSC (Meng et al., 2025) uses ViT-L/14 as the image encoder, while all our CLIP-based baselines in Table 1 use ViT-B/32. To ensure a fair comparison, we reproduce its performance with ViT-B/32 using the code provided by the author and report the best clustering results after tuning hyperparameters over $\gamma \in \{0.5, 1, 5, 10, 50, 100, 150, 300, 500\}$ and $\beta \in \{0, 100, 200, 300, 400\}$.

**Ablation study.** The intention of the first two rows of Table 2 is to analyze uni-modal counterparts of DeepMORSE. The first row is the image-only counterpart of DeepMORSE: we keep only the loss terms for the image modality, compute $C$ as $\mathcal{P}(Z_{\text{image}}^{\top} Z_{\text{image}})$ with the learned image representations, and perform spectral clustering on the affinity matrix induced by $A = |C + C^{\top}|$. The second row is the text-only counterpart of DeepMORSE: we keep only the loss terms for the text modality, compute $C$ as $\mathcal{P}(Z_{\text{text}}^{\top} Z_{\text{text}})$ with the learned text representations, and perform spectral clustering on the affinity matrix induced by $A = |C + C^{\top}|$. We report the best clustering results after tuning hyperparameters over $\gamma \in \{0.5, 1, 5, 10, 50, 100, 150, 300, 500\}$.

## C  MORE EXPERIMENTAL RESULTS

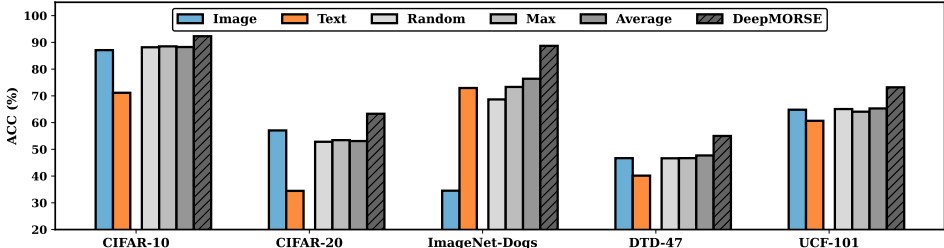

Figure C.1: **Comparison of clustering accuracy with shared versus independent self-expression.** Independent coefficients are integrated by random selection, maximum absolute values, or averaging for the final clustering result.

**The effectiveness of modality-shared self-expression.** To validate the effectiveness of using modality-shared self-expression for deep clustering, we change DeepMORSE by learning modal-dependent self-expression. Then, integrated self-expressive coefficients are computed through randomly selection, taking the maximum absolute values, or averaging the modal-dependent self-expressive coefficients. The clustering results based on these modal-dependent self-expression are illustrated in Figure C.1. Additionally, we also report the clustering results based on unimodal self-expression without multimodal integration.

As can be observed, the integration of modal-dependent self-expression yields subtle improvements comparing with utilizing only the image modality. On the contrary, our DeepMORSE consistently yields significant improvements ($8\%$ improvements for DTD-47 and UCF-101, $10\%$ improvement for ImageNet-Dogs), validating its effectiveness in uncovering modality-shared structures.

**Mechanism of modality-shared self-expression.** Let $C_{\text{img}}$ and $C_{\text{text}}$ denote the self-expression matrices that are learned independently for image and text modalities. In practice, both $C_{\text{img}}$ and $C_{\text{text}}$ contain correct coefficients that correspond to samples from the same class, and noise coefficients that correspond to samples from different classes. Intuitively, we believe that: correct relations tend to be shared across image and text, while noise relations are harder to exist simultaneously in both modalities. Thus, the modality-shared self-expression constraint promotes DeepMORSE to focus on coefficients that are more likely to be correct, i.e., corresponds to samples from the same class. In contrast, simply integrating two separate expressive coefficients (by averaging, max-pooling) cannot filter out the noise in each modality.

To validate this hypothesis, We compare the shared matrix $C$ learned by DeepMORSE to those obtained by integrating $C_{\text{img}}$ and $C_{\text{text}}$. As shown in Figure C.2, the coefficient matrix from Deep-MORSE exhibits a much cleaner block-diagonal structure: off-block entries (inter-class relations)

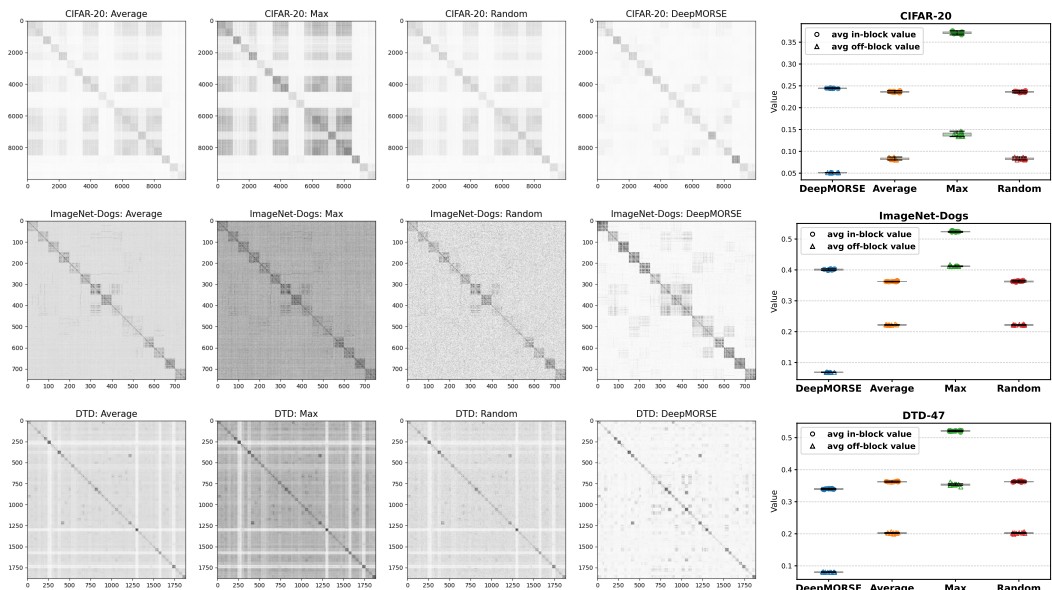

Figure C.2: **Illustration of self-expressive matrix learned by different methods.**

are strongly suppressed compared to the integrated version. Quantitatively, over 10 runs, we compute for each method: the average absolute block-diagonal value (intra-class relations), and the average absolute off-block-diagonal value (inter-class relations). The results show that the inter-class (i.e., noise) relations of DeepMORSE are consistently smaller than the integrated-coefficients baseline, and the intra-class relations of DeepMORSE are comparable. These observations validate that modality-shared coefficient encourages DeepMORSE to learn robust relations among data, and leads to a "cleaner" affinity for clustering, which contributes to the performance gap observed in the ablations.

**Generating text captions for images using MLLM.** Leveraging Multimodal Large Language Models (MLLMs) to enrich textual counterparts is appealing, especially when WordNet (which we currently use) offers limited semantic coverage. In our study, we take Qwen-VL-Plus as a representative of MLLMs.

We empirically explored two implementations: **a) Directly taking MLLM-generated captions as text counterparts.** For each image, we prompt Qwen-VL-Plus to generate a descriptive caption. We then wrap this caption into a CLIP-style prompt (e.g., "a photo of [caption]") and encode it with the pretrained CLIP text encoder to obtain textual counterparts. **b) using MLLM-generated captions to build a dataset-specific "mini-WordNet".** Alternatively, we collect captions for all images in a dataset and regard the resulting caption set as a dataset-specific, semantically compact dictionary—a "mini-WordNet". We then generate text counterparts from this dictionary and train DeepMORSE exactly as in the original WordNet-based setting.

After repeating each experiment with 5 random seeds, we report the mean clustering accuracy in Table C.3. As shown, using MLLM-generated captions yields performance comparable to or even better than the original WordNet-based setup, especially on the DTD-47 datasets, which contain texture images that are beyond the description of WordNet. However, for CIFAR-10 and CIFAR-20, the MLLM frequently misclassifies images because their resolution is very low ($32 \times 32$), making reliable caption generation difficult.

**Necessity of learnable transformations.** To validate the necessity of involving learnable transformations $f, g$, we conducted an ablation study with three variants: **a) Without $f, g$:** the self-expressive model is learned directly on the frozen CLIP features. **b) Linear $f, g$:** $f, g$ are replaced by learnable linear projections. **c) Nonlinear $f, g$:** our full model with deep nonlinear transformations. For each variant, we tune $\gamma \in \{0.5, 1, 5, 10, 50, 100, 150, 300, 500\}$ and report the best mean clustering accuracy after repeating experiments with 5 random seeds.

Table C.3: **DeepMORSE with MLLM for text counterpart generation.**

|  | CIFAR-10 | CIFAR-20 | ImageNet-Dogs | DTD-47 | UCF-101 |
|---|---|---|---|---|---|
| Original DeepMORSE | **92.3** | **63.3** | **88.7** | 55.0 | 73.2 |
| MLLM captions as text | $77.5 \pm 0.4$ | $50.3 \pm 0.6$ | $82.3 \pm 0.3$ | **56.6** $\pm 0.7$ | **74.0** $\pm 0.6$ |
| MLLM captions as miniWordNet | $92.0 \pm 0.1$ | $61.0 \pm 0.9$ | $83.6 \pm 0.5$ | $55.5 \pm 0.8$ | $73.9 \pm 0.3$ |

As can be observed in Table C.4, the performance without $f, g$ drops noticeably compared to Deep-MORSE, confirming that learned transformation is indeed necessary. Interestingly, this variant still outperforms "CLIP ($k$-means)", indicating that the raw features are better approximated by a union of subspaces than by centroids. With linear transformation, performance improves substantially over the "without $f, g$" case, indicating that even a linear projection can approximately transform data onto a union of subspaces, thereby enhancing the clustering performance. With nonlinear transformation (DeepMORSE), we obtain the best performance across all datasets, supporting our original design choice of using deep nonlinear transformations to more effectively linearize and separate the subspaces.

Table C.4: **Necessity of involving learnable transformations.** We evaluate DeepMORSE without transformation, with linear transformation, and with nonlinear transformation.

|  | CIFAR-10 | CIFAR-20 | Dogs-15 | DTD-47 | UCF-101 |
|---|---|---|---|---|---|
| w/o $f, g$ | $78.2 \pm 0.2$ | $53.0 \pm 1.0$ | $61.7 \pm 1.9$ | $45.8 \pm 0.9$ | $60.3 \pm 0.4$ |
| w/ linear $f, g$ | $92.1 \pm 0.1$ | $59.2 \pm 0.9$ | $82.3 \pm 0.6$ | $54.4 \pm 0.3$ | $70.8 \pm 0.5$ |
| DeepMORSE | **92.3** $\pm 0.1$ | **63.3** $\pm 0.6$ | **88.7** $\pm 0.5$ | **55.0** $\pm 0.4$ | **73.2** $\pm 0.5$ |

**Compared to multi-view clustering.** To thoroughly evaluate DeepMORSE in the context of multi-view clustering, we conduct the following experiments: **a) State-of-the-art multi-view clustering on multimodal datasets.** We first treat the CLIP image and text embeddings as two views, then apply state-of-the-art multi-view clustering methods, CANDY (Guo et al., 2024) and COPER (Eisenberg et al., 2025), to our multimodal data (after text generation). For fairness, we keep network architectures the same and repeat all experiments with 5 random seeds, reporting the mean performance in Table C.5. **b) DeepMORSE on multi-view benchmarks.** We then evaluate DeepMORSE on several multi-view clustering datasets (Scene15, LandUse21, NUS-WIDE), treating their multiple views as separate modalities within our framework.

Across all datasets, DeepMORSE consistently outperforms the multi-view baseline, implying that modeling modality-specific UoS structure with modality-shared relationship, as done in Deep-MORSE, provides a more effective way to leverage both multimodal and multi-view information.

Table C.5: **Comparing DeepMORSE to multi-view clustering baselines.**

|  | ImageNet-Dogs | | | DTD-47 | | | UCF101 | | | Scene15 | | | LandUse21 | | | NUS-WIDE | | |
|---|---|---|---|---|---|---|---|---|---|---|---|---|---|---|---|---|---|---|
|  | ACC | NMI | ARI | ACC | NMI | ARI | ACC | NMI | ARI | ACC | NMI | ARI | ACC | NMI | ARI | ACC | NMI | ARI |
| CANDY (Guo et al., 2024) | 81.6 | 80.4 | 71.1 | 51.5 | 62.5 | 35.5 | 56.2 | 77.0 | 47.4 | 42.0 | 41.6 | 24.7 | 30.6 | 36.5 | 16.2 | 62.1 | 49.0 | 37.0 |
| COPER (Eisenberg et al., 2025) | 82.0 | 80.8 | 72.2 | 50.1 | 61.3 | 33.8 | 56.5 | 77.8 | 47.6 | 40.7 | 42.0 | 25.0 | 31.0 | 35.9 | 16.1 | 62.1 | 49.3 | 37.7 |
| Ours | **88.7** | **84.4** | **78.3** | **55.0** | **64.9** | **39.3** | **73.2** | **86.1** | **66.7** | **43.2** | **46.2** | **28.0** | **31.4** | **37.9** | **17.4** | **63.4** | **50.6** | **43.2** |

**Clustering performance on all benchmarks.** We report the clustering performance of Deep-MORSE on all benchmarks in Table C.6, including STL-10 and ImageNet-10, where performance is nearly saturated. We also include the standard deviations, computed over five runs with different random seeds.

**Complexity analysis.** By the commutative property: $\log \det(\boldsymbol{I} + \boldsymbol{Z}\boldsymbol{Z}^\top) = \log \det(\boldsymbol{I} + \boldsymbol{Z}^\top \boldsymbol{Z})$ (see Ma et al., 2007), we reduce the computation of $\log \det(\cdot)$ from an $N \times N$ matrix to a $d \times d$ matrix. This decouples the complexity from the number of samples $N$ and yields a time and space complexity of $\mathcal{O}(d^3)$, where typically $d \ll N$. Since the computation of $\log \det(\cdot)$ is the bottleneck in optimizing DeepMORSE, this property makes the loss evaluation substantially more efficient.

**Running time comparison.** To evaluate the computational efficiency of DeepMORSE, we measure both time and memory cost on a single NVIDIA RTX 3080 GPU and an AMD Ryzen 9 5900HX

Table C.6: **Image clustering performance on all benchmarks.** The best results are in **bold** and the second best results are underlined.

| | Backbone | Text Modality | CIFAR-10 ACC | CIFAR-10 NMI | CIFAR-20 ACC | CIFAR-20 NMI | Dogs-15 ACC | Dogs-15 NMI | DTD-47 ACC | DTD-47 NMI | UCF-101 ACC | UCF-101 NMI | STL-10 ACC | STL-10 NMI | ImgNet-10 ACC | ImgNet-10 NMI |
|---|---|---|---|---|---|---|---|---|---|---|---|---|---|---|---|---|
| GCC | ResNet-18 | ✗ | 85.6 | 76.4 | 47.2 | 47.2 | 52.6 | 49.0 | - | - | - | - | 78.8 | 68.4 | 90.1 | 84.2 |
| NNM | ResNet-18 | ✗ | 83.7 | 73.7 | 45.9 | 48.0 | 58.6 | 60.4 | - | - | - | - | 76.8 | 66.3 | - | - |
| SCAN | ResNet-18 | ✗ | 88.3 | 79.7 | 50.7 | 48.6 | 59.3 | 61.2 | 46.4 | 59.4 | 61.1 | 79.7 | 80.9 | 69.8 | - | - |
| CoKe | ResNet-18 | ✗ | 85.7 | 76.6 | 49.7 | 49.1 | - | - | - | - | - | - | - | - | - | - |
| SeCu | ResNet-18 | ✗ | **93.0** | **86.1** | 55.2 | 55.1 | - | - | - | - | - | - | 83.6 | 73.3 | - | - |
| PRO-DSC | ViT-B/32 | ✗ | 86.7 | 79.1 | 58.8 | 62.1 | 32.3 | 25.8 | 48.1 | 57.0 | 68.0 | 83.6 | **98.1** | 95.4 | **99.0** | 98.0 |
| CLIP ($k$-means) | ViT-B/32 | ✗ | 74.2 | 70.3 | 45.5 | 49.9 | 38.1 | 39.8 | 42.6 | 57.3 | 58.2 | 79.5 | 94.3 | 91.7 | 98.2 | 96.9 |
| SIC | ViT-B/32 | ✓ | 92.6 | 84.7 | 58.3 | 59.3 | 69.7 | 69.0 | 45.9 | 59.6 | 61.9 | 81.0 | 98.1 | 95.3 | 98.2 | 97.0 |
| TAC | ViT-B/32 | ✓ | 91.9 | 83.3 | 60.7 | 61.1 | 83.0 | 80.6 | 50.1 | 62.1 | 68.7 | 82.3 | **98.2** | **95.5** | **99.2** | **98.5** |
| Our DeepMORSE | ViT-B/32 | ✓ | 92.3±0.1 | 84.1±0.2 | **63.3**±0.6 | **63.1**±0.5 | **88.7**±0.5 | **84.4**±0.8 | **55.0**±0.4 | **64.9**±0.4 | **73.2**±0.5 | **86.1**±0.2 | **98.2**±0.1 | **95.5**±0.3 | **99.2**±0.3 | 98.3±0.7 |

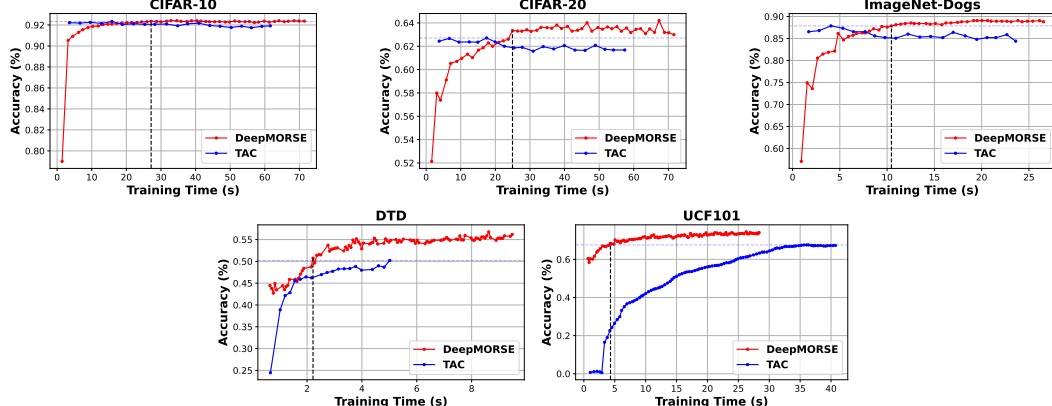

Figure C.3: **Clustering accuracy of DeepMORSE and TAC at different training time.** Red: DeepMORSE; blue: TAC.

CPU. Specifically, we plot the clustering accuracy as a function of training time for both TAC and DeepMORSE in Figure C.3. We also mark the time at which DeepMORSE reaches TAC's best performance. As shown, to reach the same clustering accuracy as TAC, DeepMORSE requires only about half of TAC's training time (and less than 30 seconds for each dataset). Furthermore, as shown in Table C.7, the time cost per epoch and the memory consumption of DeepMORSE are consistently lower. This advantage arises because TAC needs to load the neighbors of each sample into the network together with the mini-batch, which increases both memory usage and per-epoch computation.

Table C.7: **Time (s) and memory (MB) cost comparison on ImageNet-Dogs.**

| | Train↓ | Test↓ | Epoch | Memory↓ | ACC↑ |
|---|---|---|---|---|---|
| TAC | 35.09 | 0.01 | 20 | 12.42 | 83.0 |
| DeepMORSE | 40.38 | 0.42 | 50 | 3.07 | 88.7 |

**Extending beyond vision-language data.** While our current experiments focus on vision-language data, the DeepMORSE framework itself is modality-agnostic. Specifically, DeepMORSE only requires that each modality be mapped into a shared semantic feature space (as in CLIP). Given multimodal input features, we can learn a transformation for each modality (e.g., image, text, acoustics, hyperspectral) via separate encoders, and construct a modality-shared self-expressive matrix without any architectural changes. In this sense, DeepMORSE can be straightforwardly extended to additional modalities once suitable multimodal features are available.

To demonstrate this, we additionally conduct experiments with Wav2CLIP (Wu et al., 2022), which provides a joint image-text-audio embedding space. Concretely, we first generate audio counterparts for each image via multimodal sparse coding (as in Sec. 3.2) from the audio features of the FSD50K dataset (Fonseca et al., 2022), which contains 51,197 Freesound clips across 200 classes. For training DeepMORSE, we transform the audio data $u$ into representations with another learnable mapping $z_{\text{audio}} = h(u; \phi_u)$ followed by normalization. We then construct a modality-mixed representa-

tion: $z_{\mathrm{mix}} = 0.5(z_{\mathrm{img}} + z_{\mathrm{audio}})$ for image-audio; and $z_{\mathrm{mix}} = 0.45 \times z_{\mathrm{img}} + 0.45 \times z_{\mathrm{text}} + 0.1 \times z_{\mathrm{audio}}$ for image-text-audio. The modality-shared self-expressive matrix is still computed by Eq. (9), and we add the audio loss term $\gamma \|Z_{\mathrm{audio}} - Z_{\mathrm{audio}}C\|_F^2 - \rho(Z_{\mathrm{audio}})$ into the overall objective. The results are shown in Table C.8:

Table C.8: **Extending DeepMORSE beyond vision-language data.** We conduct experiments with Wav2CLIP as the pretrained tri-modal architecture.

| | CIFAR-10 | | CIFAR-20 | | ImageNet-Dogs | | DTD-47 | | UCF-101 | |
|---|---|---|---|---|---|---|---|---|---|---|
| | ACC | NMI | ACC | NMI | ACC | NMI | ACC | NMI | ACC | NMI |
| Image+text | 92.3 | 84.1 | 63.3 | **63.1** | **88.7** | 84.4 | **55.0** | **64.9** | **73.2** | **86.1** |
| Image+audio | 89.0 | 80.5 | 57.1 | 60.6 | 30.6 | 27.0 | 44.6 | 54.1 | 67.4 | 83.3 |
| Image+text+audio | **92.6** | **84.4** | **63.7** | **63.1** | **88.7** | **84.5** | 54.9 | 64.7 | 72.6 | 85.9 |

We observe a notable performance drop for the Image+audio combination, which is consistent with the findings in AudioCLIP (Guzhov et al., 2022), where ImageNet top-1 accuracy drops from 40.5% to 21.8%. This degradation is largely due to the fact that existing tri-modal encoders are trained on much smaller audio corpora (e.g., AudioSet with 2M clips) compared to the large-scale image-text data used for CLIP (400M pairs), resulting in weaker image representations.

For the Image+text+audio combination, the performance on CIFAR-10, CIFAR-20, and ImageNet-Dogs slightly improves or remains on par with the Image+text setting, demonstrating that Deep-MORSE can incorporate an additional audio modality without harming performance and sometimes with small gains. For DTD-47 and UCF-101, audio is unsurprisingly less helpful: textures (e.g., striped, grid) and many actions (e.g., applying eye makeup, playing yo-yo) are essentially silent.

$t$-**SNE visualization.** To visualize the learned representations of DeepMORSE, we present the $t$-SNE result of CLIP embeddings and our learned representations in Figure C.4. As can be observed, the raw CLIP image embeddings are not well separated, which is expected given the fine-grained categories of the datasets. In contrast, DeepMORSE effectively integrates textual information by learning modality-shared subspace structures, and consequently obtains representations that are intra-class compact and inter-classes separable.

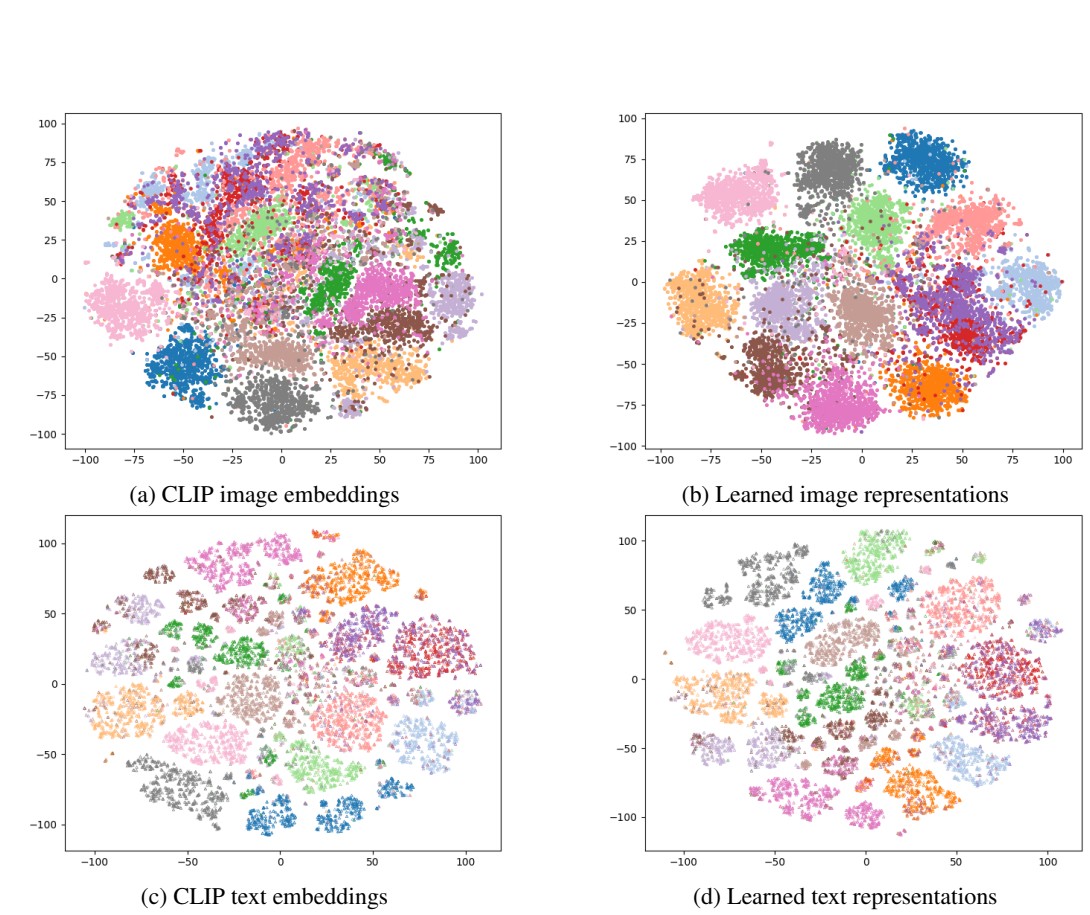

(a) CLIP image embeddings

(b) Learned image representations

(c) CLIP text embeddings

(d) Learned text representations

Figure C.4: $t$-**SNE visualization of CLIP embeddings and representations of DeepMORSE on ImageNet-Dogs dataset.** Top: image modality; bottom: text modality.

