# OpenReview forum: "Learning Deep Modality-Shared Self-Expressiveness for Image Clustering with Textual Information"
_ICLR.cc/2026/Conference — Submitted to ICLR 2026_

### Official Review · Reviewer_xkPq · 2025-10-25

**Soundness:** 3
**Presentation:** 3
**Contribution:** 3
**Rating:** 4
**Confidence:** 4

**Summary:**

This paper proposes a deep modality-shared self-expressive model for multi-modal image clustering which can simultaneously learns structured representations conforming to the union of modality-specific subspaces and discovers structures shared across modalities. Experiments on image clustering benchmarks and several downstream tasks demonstrate the effectiveness of proposed method.

**Strengths:**

1. This paper proposes a simple-yet-effective method for multi-modal image clustering, which relies on a deep modality-shared self-expressive model.
2. The proposed model can jointly learn representations conforming to a union of modality-specific subspaces and discovers shared structures across modalities.
3. The learned structured representations can be directly applied to downstream tasks including image retrieval and zero-shot classification.

**Weaknesses:**

1. The motivation for this work requires clarification. The paper repeatedly notes that "the distribution of the aligned representation in existing methods remains unclear," but the term "unclear" is not sufficiently defined. It would be helpful to illustrate this limitation more concretely, for instance, by providing visualizations on toy examples. Furthermore, the rationale behind why the proposed deep modality-shared self-expressive model can make sense remains unclear. It’s better to provide intuitive explanations or theoretical/experimental analysis to establish its foundation.
2. Regarding Table 1, several details require clarification. First, the distinction between "CLIP (k-means)" and "CLIP (zero-shot)" is unclear. Why the former is without textual information and the latter is with textual information? The authors should provide the details about these two settings in the paper. Second, the superscript attached to "PRO-DSC" is undefined—please clarify if this denotes some missing information or is simply a typographical error.
3. Regarding the second row of Table 2, the configuration is described as utilizing only the textual modality. However, the corresponding task is image clustering. Does this imply that image clustering is performed directly using the image representations from the pre-trained CLIP model, without any fine-tuning?
4. Regarding Figure 4, the analysis of hyperparameter sensitivity, which is currently conducted on only two datasets, may not be sufficient to draw general conclusions about the model's robustness. To provide a more comprehensive evaluation, it is essential to extend this analysis to include all datasets used in the study, as different datasets may exhibit varying sensitivities to hyperparameter changes.
5. For zero-shot classification, the authors utilize the known categories of the datasets to construct the dictionary, which contains the embeddings of prompts “A photo of class” extracted by a pretrained CLIP text encoder. And the image embeddings are also sourced from pretrained CLIP model. When using Eq. (11) to obtain the textual counterpart for each image embedding, the “zero-shot” scenario seems to be broken since the model gains prior knowledge of the test categories.
6. How about generating text captions for the images using MLLMs and then encode the image/caption pairs into embedded space using CLIP?

**Questions:**

See Weaknesses.

---

> ### Author Response · Authors · 2025-11-22
> **Author responses to Reviewer xkPq (Part 1)**
>
> We thank the reviewer for the valuable time and effort in reviewing our
> paper and we are grateful that the reviewer recognizes our work as
> "simple-yet-effective".
>
> ## **W1: Regarding the motivation.**
>
> > *The motivation for this work requires clarification. The paper repeatedly notes that "the distribution of the aligned representation in existing methods remains unclear," but the term "unclear" is not sufficiently defined. It would be helpful to illustrate this limitation more concretely, for instance, by providing visualizations on toy examples. Furthermore, the rationale behind why the proposed deep modality-shared self-expressive model can make sense remains unclear. It’s better to provide intuitive explanations or theoretical/experimental analysis to establish its foundation.*
>
> ### **1) What we mean by "the distribution \... remains unclear."**
>
> We note here that ``the distribution \... remains unclear'' refers to that the **geometric structure** of the aligned representations (e.g., a mixture of Gaussians, a union of subspaces, a union of manifolds) is not clearly or explicitly specified in previous work. In contrast, our DeepMORSE is explicitly designed to learn the modality-specific representations that form a union-of-subspaces (UoS). As justified in (Meng et al., ICLR 2025), a deep self-expressive model with a $\log\det(\cdot)$ regularization encourages the learned representations lying on a union of subspaces under mild conditions. In the revised manuscript, we have revised the relevant description and modified Figure 1 to clearly illustrate what we mean by "unclear".
>
> ### **2) Empirical evidence**
>
> Following the reviewer's constructive suggestion, we have added visualizations of the learned representations of TAC (Li et al., ICML 2024) and our DeepMORSE via PCA on each dataset (see Fig. 5). The reason for using a linear projection via PCA is that it can preserve the global linear structure of the learned representations. For our DeepMORSE (see rows 3 and 4 in Fig. 5), the projected embeddings form approximately a union of orthogonal low-dimensional subspaces, where the representations of each class are well aligned with a subspace. For TAC (see rows 1 and 2 in Fig. 5), the projected embeddings exhibit a variety of structures that cannot be easily approximated by a union of centroids, subspaces, or even manifolds.
>
> ### **3) Rationale behind DeepMORSE**
>
> The crucial point of our DeepMORSE is not to align all modalities into a single shared space, but to preserve the modality-specific structures and extract what is shared across modalities.
>
> Concretely, our DeepMORSE achieves this by:
>
> - **Introducing deep self-expressive models for each modality** where each modality is encouraged to learn its own latent space in which the representations form a modality-specific UoS;
>
> - **Using modality-shared self-expressive coefficients** to capture the modality-invariant relationship, by which robust clusters can be discovered.
>
> [1] Meng et al., \"Exploring a Principled Framework for Deep Subspace
> Clustering\", ICLR 2025.
>
> [2] Li et al., \"Image Clustering with External Guidance\", ICML 2024.

---

> ### Author Response · Authors · 2025-11-22
> **Author responses to Reviewer xkPq (Part 2)**
>
> ## **W2 / W5: Clarifications on zero-shot setting.**
>
> > *Regarding Table 1, several details require clarification. First, the distinction between "CLIP (k-means)" and "CLIP (zero-shot)" is unclear. Why the former is without textual information and the latter is with textual information? The authors should provide the details about these two settings in the paper.*
>
> > *For zero-shot classification, the authors utilize the known categories of the datasets to construct the dictionary, which contains the embeddings of prompts “A photo of class” extracted by a pretrained CLIP text encoder. And the image embeddings are also sourced from pretrained CLIP model. When using Eq. (11) to obtain the textual counterpart for each image embedding, the “zero-shot” scenario seems to be broken since the model gains prior knowledge of the test categories.*
>
> ### **1) Distinction between "CLIP ($k$-means)" and "CLIP (zero-shot)"**
>
> - **CLIP ($k$-means) - without textual information.** Here, we use only
>   the CLIP image encoder as a feature extractor. For each dataset, we
>   extract image embeddings with the pretrained CLIP image encoder and
>   then directly run the $k$-means algorithm on these embeddings. There are no class
>   names, prompts, or text encoder being used. This is therefore a purely
>   image-based clustering baseline without textual information.
>
> - **CLIP (zero-shot) - with textual information.** In contrast, "CLIP
>   (zero-shot)" follows the **standard zero-shot protocol** introduced in
>   CLIP (Radford et al., ICML 2021): for each dataset, we construct
>   textual prompts such as "a photo of \[class\]", encode them with the
>   CLIP text encoder, and classify each image by comparing its CLIP image
>   embedding to all prompt embeddings and taking the most similar one.
>   This baseline explicitly uses textual information (i.e., class names) via
>   the text encoder.
>
> To avoid confusion and to keep Table 1 focused on clustering methods, in
> the revised manuscript we remove the "CLIP (zero-shot)" row from Table 1
> and keep "CLIP ($k$-means)" as the unsupervised CLIP baseline, while
> clearly explaining the standard CLIP zero-shot protocol in Sec. 4.3
> where we discuss downstream evaluations.
>
> ### **2) Prior knowledge of test categories in zero-shot classification.**
>
> We understand the reviewer's concern that using prior knowledge of test
> categories to build the text dictionary might seem to give our DeepMORSE an
> advantage over other baselines. Here we clarify that this is exactly the
> standard setting of zero-shot classification (where the label set is known) and thus is fair for all the baselines in Table 4.
>
> [1] Radford et al., \"Learning Transferable Visual Models From Natural
> Language Supervision\", ICML 2021.
>
> ## **W2: Regarding the superscript attached to "PRO-DSC".**
>
> > *Second, the superscript attached to "PRO-DSC" is undefined—please clarify if this denotes some missing information or is simply a typographical error.*
>
> We thank the reviewer for pointing out that the superscript on "PRO-DSC"
> was not explained. This superscript indicates that the reported results
> are **reproduced** under our experimental setup.
>
> Specifically, PRO-DSC (Meng et al., ICLR 2025) uses ViT-L/14 as the
> image encoder, whereas all other CLIP-based baselines in Table 1 use
> ViT-B/32. To ensure a fair comparison with a consistent backbone, we
> re-implemented PRO-DSC with ViT-B/32 and denote these reproduced results
> with a superscript. The hyper-parameters are tuned over
> $\gamma\in\\{0.5,1,5,10,50,100,150,300,500\\},\beta\in\\{0,100,200,300,400\\}$
> to report the best performance. We have added an explicit explanation of
> the superscript and the corresponding hyper-parameter ranges in the
> experimental setup of the revised manuscript.
>
> [1] Meng et al., \"Exploring a Principled Framework for Deep Subspace
> Clustering\", ICLR 2025.

---

> ### Author Response · Authors · 2025-11-22
> **Author responses to Reviewer xkPq (Part 3)**
>
> ## **W3: Regarding the second row of Table 2.**
>
> > *Regarding the second row of Table 2, the configuration is described as utilizing only the textual modality. However, the corresponding task is image clustering. Does this imply that image clustering is performed directly using the image representations from the pre-trained CLIP model, without any fine-tuning?*
>
> We thank the reviewer for pointing out that our description of this
> experiment is too coarse and may lead to confusion.
>
> The results in the first two rows of Table 2 are intended to serve as unimodal counterparts of our DeepMORSE used for ablation. The results in the second row are obtained by running the **text-only** version of our DeepMORSE, i.e., we keep only the loss terms for the text modality, compute text-modal expressive matrix $\boldsymbol{C}$ by $\mathcal{P}(\boldsymbol{Z}\_\text{text}^\top\boldsymbol{Z}\_\text{text})$ with the learned text representations, and perform spectral clustering on the affinity matrix induced by $\boldsymbol{A}=|\boldsymbol{C}+\boldsymbol{C}^\top|$. The clustering assignments are evaluated against the ground-truth labels of the paired images, since each image-text pair shares the same semantic label.
>
> Thus, this setting **does not** imply that image clustering is performed directly on the frozen CLIP image embeddings. Instead, we train a unimodel (i.e., text-only) counterpart of our DeepMORSE and report its clustering performance.
>
> To avoid such confusion, we have added the corresponding experimental details in Appendix B of the revised manuscript.
>
> ## **W4: Regarding the sensitivity.**
>
> > *Regarding Figure 4, the analysis of hyperparameter sensitivity, which is currently conducted on only two datasets, may not be sufficient to draw general conclusions about the model's robustness. To provide a more comprehensive evaluation, it is essential to extend this analysis to include all datasets used in the study, as different datasets may exhibit varying sensitivities to hyperparameter changes.*
>
> We thank the reviewer for this careful suggestion and agree that examining hyper-parameter sensitivity on only two datasets is limited. In the revised manuscript, we have extended the sensitivity study to all image clustering benchmarks used in this work (see Figure 4).
>
> To make the robustness assessment more convincing and statistically sound, we also repeat each experiment for 5 trials with different random seeds and report the average performance together with the standard deviation (shown as shaded areas around the curves). The extended results will provide a more comprehensive view of how our DeepMORSE behaves across datasets with varying hyper-parameters.

---

> ### Author Response · Authors · 2025-11-22
> **Author responses to Reviewer xkPq (Part 4)**
>
> ## **W6: Generating text captions for images using MLLM.**
>
> > *How about generating text captions for the images using MLLMs and then encode the image/caption pairs into embedded space using CLIP?*
>
> We thank the reviewer for this very insightful suggestion. Leveraging MLLMs to enrich textual counterparts is indeed quite appealing, especially because WordNet (which we currently use) offers only limited semantic coverage. In our study, we attempt to take Qwen-VL-Plus as a representative MLLM.
>
> We empirically explored **two implementations**:
>
> 1.  **Directly using MLLM-generated captions as text counterparts.** For
>     each image, we prompt Qwen-VL-Plus to generate a descriptive
>     caption. We then wrap this caption into a CLIP-style prompt (e.g.,
>     "a photo of \[caption\]") and encode it with the pretrained CLIP
>     text encoder to obtain textual counterparts.
>
> 2.  **Using MLLM-generated captions to build a dataset-specific
>     "mini-WordNet".** Alternatively, we collect captions for all images
>     in a dataset and regard the resulting caption set as a
>     dataset-specific, semantically compact dictionary---a
>     "mini-WordNet". We then generate text counterparts from this
>     dictionary and train our DeepMORSE exactly as in the original
>     WordNet-based setting.
>
> We run each experiment for 5 trials with different random seeds and report the average clustering accuracy in the table below.
>
> | Method                        | CIFAR-10 | CIFAR-20 | Dogs-15 | DTD-47        | UCF-101        |
> |------------------------------|---------:|---------:|--------------:|--------------:|---------------:|
> | Original DeepMORSE           | **92.3**±0.1 | **63.3**±0.6 | **88.7**±0.5      | 55.0±0.4          | 73.2±0.5           |
> | MLLM captions as text        | 77.5±0.4 | 50.3±0.6 | 82.3±0.3      | **56.6±0.7**  | **74.0±0.6**   |
> | MLLM captions as miniWordNet | 92.0±0.1 | 61.0±0.9 | 83.6±0.5      | 55.5±0.8      | 73.9±0.3       |
>
>
>
> These results show that:
>
> 1.  Directly using MLLM captions as text counterparts can hurt
>     performance on low-resolution datasets such as CIFAR-10 and
>     CIFAR-20, where $32\times32$ images make reliable caption generation
>     difficult; the MLLM often misinterprets the content.
>
> 2.  For texture-like data (DTD-47) and action data (UCF-101), MLLM
>     captions can be beneficial, probably because they capture semantic
>     attributes beyond WordNet's coverage.
>
> 3.  The "mini-WordNet" strategy yields performance that is comparable to
>     the original WordNet-based DeepMORSE on most datasets, and clearly
>     more stable than directly using raw captions.
>
> Overall, we regard MLLM-assisted text counterpart generation as a
> promising research direction. It can enrich or even replace WordNet when we carefully post-process the captions. We thank the reviewer again for
> this intriguing idea and have included these experiments and
> observations in the revised manuscript (Appendix C).

---

> ### Author Response · Authors · 2025-12-02
>
> We appreciate the insightful comments and constructive suggestions from reviewer xkPq. Due to approaching the end of the rebuttal period, we kindly invite the reviewer to take a few minutes to read our responses in the rebuttal, to see if there are still unclear issues to further clarify. Thanks!

---

### Official Review · Reviewer_4ETM · 2025-10-29

**Soundness:** 2
**Presentation:** 2
**Contribution:** 2
**Rating:** 4
**Confidence:** 4

**Summary:**

This paper addresses the challenge of cross-modal retrieval, where the goal is to retrieve relevant samples across different modalities (e.g., retrieving images given text queries, or vice versa). The authors propose a modality-specific deep learning framework that explicitly learns separate but aligned representations for each modality.

**Strengths:**

1. Experimental results show consistent improvements over strong baselines (e.g., DCCA, Corr-AE, CCA) across multiple datasets, suggesting the proposed approach generalizes well.

2. The paper clearly identifies the limitations of enforcing overly tight shared embedding spaces. The motivation for learning modality-specific representations is intuitive.

**Weaknesses:**

1. While the concept of modality-specific embeddings is valuable, the implementation primarily extends known ideas (e.g., DCCA) rather than introducing a fundamentally new network design.

2. The paper could benefit from a deeper theoretical justification for the chosen balance between intra- and inter-modal losses. The trade-off parameter is empirically chosen without clear reasoning.

3. It’s unclear how much each component (e.g., intra-modal loss, modality-specific subnetworks) contributes to the final performance. A comprehensive ablation table would strengthen the claims.

4. The two-stream modality-specific design likely doubles training cost, but the paper doesn’t quantify this or discuss efficiency trade-offs.

**Questions:**

1. How sensitive is the retrieval performance to the weighting between intra- and inter-modal losses? Is there a principled way to select this parameter?

2. Can this approach handle large-scale multimodal datasets (e.g., millions of image–text pairs) without significant computational overhead?

3. Does the method use explicit negative sampling or rely entirely on pairwise constraints? Could incorporating contrastive loss improve robustness?

4. Could the same framework extend naturally to more than two modalities (e.g., audio–video–text)?

---

> ### Author Response · Authors · 2025-11-22
> **Author responses to Reviewer 4ETM (Part 1)**
>
> We thank the reviewer for the time and the feedbacks. However, we found that several key remarks in the review comments appeared to be based on misunderstandings of our problem setting, loss function, and experiments. To make it clear, we first briefly highlight the main clarifications and then reply the questions (or issues) point-to-point in detail.
>
> ## **Main Clarifications.**
>
> - **Misunderstanding in our main task**. Our primary task is to address a text-assisted image **clustering**, rather than a cross-modal **retrieval**, which is merely a downstream evaluation to test the transferability of the learned representations by our proposed approach.
>
> - **Mistakes in mentioning baselines**. In the first items of "Strength" and "Weaknesses" in the review comments, it is mentioned of that we compared against "strong baselines (e.g., DCCA, Corr-AE, CCA)". However, none of the mentioned algorithms, e.g., DCCA, Corr-AE, and CCA, are used as baselines in our submission.
> Our baseline algorithms are GCC, NNM, SCAN, PRO-DSC, SIC, and TAC, which are standard unimodal/multimodal clustering algorithms.
>
> - **Misunderstanding in loss design**. In the review comments, it is repeatedly referred to "intra- and inter-modal losses". However, our objective function contains **only intra-modal self-expressive terms** plus coding-rate regularizers, where cross-modal interaction is captured via a shared self-expressive matrix, not via a separate inter-modal loss.
>
> - **Mistakes in the remarks on ablations, sensitivity and efficiency**. In the review comments, it is claimed that the contribution of each component is unclear, the sensitivity of performance to hyper-parameter is not investigated, and that efficiency trade-offs are not discussed. In fact, we have provided a set of ablation studies (in Table 2), sensitivity analysis (in Figure 4), and time/memory analysis (in Table C.7) in the original submission.
>
> Now, we are ready to elaborate on these points while answering the reviewer's specific comments point-to-point.
>
> ## **W1: Regarding the comparison to DCCA.**
>
> > *While the concept of modality-specific embeddings is valuable, the implementation primarily extends known ideas (e.g., DCCA) rather than introducing a fundamentally new network design.*
>
> The goals of our DeepMORSE and DCCA are NOT to introduce a fundamentally new backbone architecture, but to propose a new objective function (i.e., framework) for multimodal representations learning. The motivation of our DeepMORSE in learning representations is different from that of DCCA. To be specific:
>
> - Our DeepMORSE aims to learn modality-specific representations that align with union of modality-specific subspaces, coupled through shared self-expressive coefficients; whereas
> - DCCA is designed to learn maximally correlated representations of different views.
>
> Thus, although the encoders themselves are standard modules (the same is also in DCCA), the core contribution of our DeepMORSE is fundamentally different from the correlation-based approaches such as DCCA.
>
> [1] Andrew et al., \"Deep Canonical Correlation Analysis\", ICML 2013.
>
> ## **W2 / Q1: Regarding the hyper-parameter.**
>
> > *The paper could benefit from a deeper theoretical justification for the chosen balance between intra- and inter-modal losses. The trade-off parameter is empirically chosen without clear reasoning.*
>
> > *How sensitive is the retrieval performance to the weighting between intra- and inter-modal losses? Is there a principled way to select this parameter?*
>
> We appreciate the reviewer's comment, but this point stems from a misunderstanding of our objective function.
>
> ### **1) We do not use an inter-modal loss.**
>
> Our loss in Eq. (10) is a sum of intra-modality terms for image and text, respectively. For each modality, the loss contains a deep self-expressive term, and a coding-rate regularizer. These two terms jointly encourage the representations to form a union-of-subspaces structure.  The cross-modality interaction is not implemented via a separate "inter-modal loss"; instead, it arises from the constraint that both modalities share the same self-expressive coefficient matrix.
>
> ### **2) The hyper-parameter $\gamma$ can be picked a principled way.**
>
> The trade-off hyper-parameter $\gamma$ operates within each modality: it balances the self-expressive term and the coding-rate regularizer.
> For the principle of selecting $\gamma$, please kindly refer to our response to Reviewer D4J1 or Appendix B in the revised manuscript.
>
> ## **W3: Regarding the ablation study.**
>
> > *It’s unclear how much each component (e.g., intra-modal loss, modality-specific subnetworks) contributes to the final performance. A comprehensive ablation table would strengthen the claims.*
>
> In Section 4.2 of our initial submission, a set of ablation studies have already been summarized in Table 2, where the unimodal variants, the effect of the regularization terms, and different text generation methods are evaluated.

---

> ### Author Response · Authors · 2025-11-22
> **Author responses to Reviewer 4ETM (Part 2)**
>
> ## **W4: Regarding the training cost.**
>
> > *The two-stream modality-specific design likely doubles training cost, but the paper doesn’t quantify this or discuss efficiency trade-offs.*
>
> We appreciate the reviewer's concern on the potential computational
> cost of a two-stream modality-specific design.
>
> ### **1) Why two modality-specific streams are needed?**
>
> Images and texts lie in fundamentally different feature spaces, suffering from a substantial modality gap (Liang et al.,NeurIPS 2022). Learning from both modalities with a single shared encoder typically leads to suboptimal representations for at least one modality. For this reason, all vision-language frameworks (e.g., CLIP and its variants) adopt separate encoders for image and
> text. Thus, the two-stream design is not an ad-hoc but a necessary
> choice to effectively learn with multimodal data.
>
> To verify this hypothesis, we evaluate our DeepMORSE with a single-stream transformation and report the experimental results in the table below.
>
> | Method        | CIFAR-10   | CIFAR-20   | Dogs-15    | DTD-47     | UCF-101    |
> |--------------|-----------:|-----------:|-----------:|-----------:|-----------:|
> | Single-stream| 79.6±0.3   | 55.7±0.5   | 77.7±1.1   | 51.2±0.3   | 70.1±0.7   |
> | DeepMORSE    | **92.3±0.1** | **63.3±0.6** | **88.7±0.5** | **55.0±0.4** | **73.2±0.5** |
>
> As can be read, the performance of our DeepMORSE's single-stream counterpart degrades compared to the two-stream model, which empirically validates our hypothesis.
>
> ### **2) Training cost is quantified and compared.**
>
> We have analyzed the complexity and time cost of our DeepMORSE in our initial submission and added a more thorough comparison to baselines in Figure C.3, which shows that our DeepMORSE reaches TAC's best accuracy in less training time; if more training time is allowed, it continues to yield better performance.
>
> [1] Liang et al., \"Mind the Gap: Understanding the Modality Gap in
> Multi-modal Contrastive Representation Learning\", NeurIPS 2022.
>
> ## **Q2: Regarding the large-scale datasets.**
>
> > *Can this approach handle large-scale multimodal datasets (e.g., millions of image–text pairs) without significant computational overhead?*
>
> Our DeepMORSE is able to handle large-scale datasets. Here, we also conducted experiments on ImageNet-1K, which contains over 1.28M of image data. We repeat experiments with 5 different random seeds and report the mean results (with standard deviation). As shown in the table below, our DeepMORSE still outperforms state-of-the-art baselines on this large-scale benchmark.
>
> | Method    | ACC       | NMI       | ARI       |
> |-----------|----------:|----------:|----------:|
> | SIC       | 47.0      | 77.2      | 34.3      |
> | TAC       | 58.2      | 79.9      | 43.5      |
> | DeepMORSE | **60.8±0.2** | **80.9±0.2** | **44.9±0.6** |
>
>
> ## **Q3: Regarding the contrastive loss.**
>
> > *Does the method use explicit negative sampling or rely entirely on pairwise constraints? Could incorporating contrastive loss improve robustness?*
>
> We thank the reviewer for this question. Our DeepMORSE does not use explicit negative sampling or a contrastive loss in its training objective.
> Instead, our DeepMORSE relies on intra-modal self-expressive terms together with a coding-rate regularizer. The entries of $\boldsymbol{C}$ explicitly encode pairwise relations in data, but we do not define positive/negative pairs as in contrastive learning.
>
> It is worth noting that the CLIP backbone we use as feature extractor has already been pretrained with a contrastive objective involving negative sampling, thus the initial image-text embeddings have already been benefit from such training. Our DeepMORSE then focuses on a different aspect: learning a union-of-subspaces structure with a modality-shared self-expressive matrix to obtain cleaner, more cluster-friendly multimodal representations.
>
> Besides, we argue that incorporating an additional contrastive loss on top of our
> self-expressive objective is certainly possible and may further improve the
> robustness in some settings as it is orthogonal to the main
> contribution of our work. Thus, we leave the integration of contrastive
> objectives as an interesting direction for future work.
>
> ## **Q4: Regarding the experiments with more than two modalities.**
>
> > *Could the same framework extend naturally to more than two modalities (e.g., audio–video–text)?*
>
> We appreciate the reviewer for the intriguing idea. We have extended our DeepMORSE to the image-audio-text modalities. Please kindly refer to Appendix C of the revised manuscript or our responses to Reviewer D4J1 for more details.

---

> > ### Comment · Reviewer_4ETM · 2025-11-25
> > **Correction**
> >
> > Thanks for pointing the misunderstanding out — the reviewer's wording was imprecise.
> >
> > The reviewer didn’t mean that Eq. (10) has separate, explicitly weighted intra- and inter-modal loss terms. $C$ is computed from the mixed representation $Z_{\text{mix}}$, and Eq. (10) combines the self-expression terms (using the shared $C$) with the coding-rate regularizers for each modality.
> >
> > What the reviewer really wanted to ask is:
> >
> > * Is there any practical guidance or principled rule for choosing $\gamma$ (and related hyperparameters) on a new dataset, beyond tuning it empirically?
> >
> > If the authors could comment on this trade-off, it would clarify how to use the method in practice.

---

> > > ### Author Response · Authors · 2025-11-25
> > > **Responses to the Corrections**
> > >
> > > Thanks for the reviewer 4ETM to drop replies timely and kindly ask for a further clarification.
> > >
> > > The hyper-parameter $\gamma$ is selected as follows. First of all, as justified in (Meng et al., ICLR 2025), the parameter $\gamma$ to prevent catastrophic collapse should be set linearly and be upper bounded with $\alpha=d/(N\epsilon^2)$, where $d$ is the output dimension, $N$ is the batch-size, and $\epsilon$ is the coding precision. Since that our DeepMORSE uses the same $N$ and $\epsilon$ for all experiments, we have that the parameter $\gamma$ should scale linearly with the output dimension $d$. In experiments, we set $\gamma$ approximately via a simple rule $\gamma\approx0.5 \times d + 80$, which can also be used on other datasets without manual search for hyper-parameter.
> > >
> > > Regarding the hyper-parameter, please refer to the discussions L940-966 in Appendix B.
> > >
> > > [1] Meng et al., \"Exploring a Principled Framework for Deep Subspace Clustering\", ICLR 2025.

---

> ### Author Response · Authors · 2025-12-02
>
> Thanks for reading our responses and providing feedbacks. We kindly invite the reviewer to read our responses again to see if there are still unclear issues. Thanks!

---

### Official Review · Reviewer_D4J1 · 2025-10-31

**Soundness:** 3
**Presentation:** 3
**Contribution:** 3
**Rating:** 6
**Confidence:** 4

**Summary:**

- This paper introduces the DeepMORSE to address the challenge in image clustering with textual information, where existing modality alignment methods often fail to preserve modality-specific structures and leave the overall representation distribution unclear.

- DeepMORSE operates by simultaneously learning structured representations that conform to the union of modality-specific subspace structures and explicitly discovering patterns shared across modalities via a modality-shared self-expressive model.

- In practical scenarios where text counterparts are not readily available, the approach generates necessary textual data for each image by solving a cross-modal sparse coding problem to ensure both semantic accuracy and adherence to a union-of-subspaces structure.

- Extensive experiments demonstrate that DeepMORSE achieves state-of-the-art clustering performance on seven benchmarks, observing performance improvements exceeding 4% on the UCF-101, DTD-47, and ImageNet-Dogs datasets, while also showing strong transferability to image retrieval and zero-shot classification without requiring task-specific optimization.

**Strengths:**

This paper's core strength lies in its simplicity. DeepMORSE overcomes limitations of prior alignment methods by simultaneously learning structured representations and explicitly discovering patterns shared across modalities through a modality-shared model.

The approach achieves descent clustering performance across seven benchmarks, reporting improvements in clustering accuracy, including gains exceeding on the UCF-101, DTD-47, and ImageNet-Dogs datasets.

Furthermore, the learned structured representations demonstrate transferability and robustness, achieving comparable results on downstream tasks such as image retrieval and zero-shot classification without requiring any additional optimization.

**Weaknesses:**

1. One weakness of the current framework is its limitation in scope to vision-language data, and the necessary extension of the method to other modalities, such as acoustics or hyperspectral imagery, has yet to be investigated.

2. While exhibiting low memory consumption, DeepMORSE requires a slightly longer total training and testing time compared to baselines, such as TAC. Leading to challenges in real-world application deployment.

3. The necessity of leveraging textual information necessitates a crucial pre-processing step, utilizing cross-modal sparse coding and a predefined dictionary to generate textual counterparts when image-text pairs are unavailable, thereby introducing external complexity and dependence on the quality and sparsity of this synthetic data. This unpaired data situation is very common in real-world settings. Furthermore, ablation studies reveal that DeepMORSE suffers a sharp degradation in clustering performance when components relying on the textual modality are removed, demonstrating that the overall effectiveness is highly dependent on the presence and successful integration of both modality expressions.

4. Although generally robust to hyperparameter choices, the model requires task-specific adjustments. Specifically, increasing the output dimensions and balancing the hyperparameter **($\gamma$)** for downstream evaluations on datasets containing more categories suggests that the default configuration is not universally optimal for larger class counts.

**Questions:**

1. Please answer the questions in the weakness section.

2.  The paper explicitly lists the limitation that the theoretical underpinnings of modality-shared self-expression remain largely unexplored, leaving the fundamental working mechanism insufficiently understood. Can the authors elaborate on the specific challenges encountered when attempting to derive theoretical guarantees for the shared coefficient matrix $C$ (Equation 4) in the multimodal setting, similar to those established for unimodal sparse subspace clustering? What are the most promising theoretical avenues for future research to address this gap?

3. The authors noted that for datasets with more than 128 categories (StanfordCars, SUN397), it was necessary to increase the output dimension $d$ and enlarge the balancing hyperparameter $\gamma$. Is there a principled rule or heuristic that can guide the selection of $d$ and $\gamma$ based on the number of classes $C$ or the complexity of the dataset to ensure the model maintains optimal performance and avoids convergence issues, instead of relying on manual tuning?

4.  Ablation studies show that DeepMORSE, which uses modality-shared self-expression yields significantly larger improvements than simply combining coefficients derived independently from each modality. Can the authors provide a more detailed, perhaps qualitative, explanation of why the rigid constraint of enforcing the exact same coefficient matrix $C$ across both image and text representations (in Equation 4) is crucial for uncovering robust shared structures, compared to merely integrating two separate affinity matrices?

---

> ### Author Response · Authors · 2025-11-22
> **Author responses to Reviewer D4J1 (Part 1)**
>
> We appreciate the reviewer D4J1 for reading our submission carefully, for the supportive rating, and for the insightful and constructive comments.
>
> ## **W1: On extending beyond vision-language data.**
>
> > *One weakness of the current framework is its limitation in scope to vision-language data, and the necessary extension of the method to other modalities, such as acoustics or hyperspectral imagery, has yet to be investigated.*
>
> We thank the reviewer for this constructive suggestion. The idea of extending our DeepMORSE to other modalities is indeed interesting. While our current experiments focus on vision-language data, the **DeepMORSE framework itself is modality-agnostic and thus can be flexibly extended to settings with more modalities**.
>
> Specifically, given multimodal input features, we can learn modality-specific transform for each modality (e.g., image, text, acoustics, hyperspectral) via modality-specific encoders, and then construct a modality-shared self-expressive coefficient matrix without any architectural change. In this sense, our DeepMORSE can be straightforwardly extended to scenarios where the data are provided in modalities beyond vision and language.
>
> To demonstrate this, we conduct experiments with Wav2CLIP (Wu et al., ICASSP 2022), which provides a joint image-text-audio embedding space.
>
> 1.  We first generate audio counterparts for each image via multimodal sparse coding (as in Sec. 3.2) from the audio features of the FSD50K dataset, which contains 51,197 Freesound clips across 200 classes.
>
> 2.  For training our DeepMORSE, we transform the audio data $\boldsymbol{u}$ into the representations with another learnable mapping    $\boldsymbol{z}\_\text{audio}=h(\boldsymbol{u};\boldsymbol{\phi}\_{\boldsymbol{u}})$ followed by normalization.
>
> 3.  We then construct a modality-mixed representation: $\boldsymbol{z}\_\text{mix}=0.5(\boldsymbol{z}\_\text{img}+\boldsymbol{z}\_\text{audio})$ for image-audio; and     $\boldsymbol{z}\_\text{mix}=0.45\times\boldsymbol{z}\_\text{img}+0.45\times\boldsymbol{z}\_\text{text} + 0.1\times\boldsymbol{z}\_\text{audio}$ for image-text-audio.
>
> 4.  The modality-shared self-expressive matrix is still computed by Eq. (9), and we add the audio loss term
>     $\gamma||\boldsymbol{Z}\_\text{audio}-\boldsymbol{Z}\_\text{audio}\boldsymbol{C}||\_F^2-\rho(\boldsymbol{Z}\_\text{audio})$
>     into the overall objective.
>
> The average results  (ACC\% / NMI\%) over 5 trials are shown in the table below.
>
> | Modality           | CIFAR-10  | CIFAR-20  | Dogs-15  | DTD   | UCF-101  |
> |--------------------|------:|------------:|-----------------:|----------:|---------:|
> | Image+text         | 92.3 / 84.1         | 63.3 / **63.1**     | **88.7** / 84.4              | **55.0** / **64.9**    | **73.2** / **86.1**      |
> | Image+audio        | 89.0 / 80.5         | 57.1 / 60.6         | 30.6 / 27.0              | 44.6 / 54.1        | 67.4 / 83.3          |
> | Image+text+audio   | **92.6** / **84.4**     | **63.7** / **63.1**     | **88.7** / **84.5**          | 54.9 / 64.7        | 72.6 / 85.9          |
>
>
> We observe a notable performance drop for the **Image+audio** combination, which is consistent with the findings reported in AudioCLIP (Guzhov et al., ICASSP 2022): the ImageNet top-1 classification accuracy drops from 40.5% to 21.8%. This degradation is largely due to the fact that existing tri-modal encoders are trained on much smaller audio corpora (e.g., AudioSet with 2M clips) compared to the large-scale image-text data used for CLIP (400M pairs), resulting in weaker image representations.
>
> For the **Image+text+audio** combination, the performance on CIFAR-10, CIFAR-20, and ImageNet-Dogs slightly improves or remains on par with the Image+text setting, demonstrating that our DeepMORSE can incorporate an additional audio modality without harming performance and sometimes with small gains. For DTD-47 and UCF-101, the audio data is unsurprisingly less helpful because the textures (e.g., striped, grid) and many actions (e.g., applying eye makeup, playing yo-yo) are essentially silent.
>
> Overall, these empirical results show that **our DeepMORSE can be extended to image-text-audio data**. However, consistent with the observations in (Guzhov et al., ICASSP 2022), the clustering performance might not necessarily exceed that of the image-text setting. This depends upon whether the data with more modalities provide sufficient information to recover the sharing distribution structures across modalities. We have added this discussion and the above results to the revised manuscript.
>
> [1] Wu et al., \"Wav2CLIP: Learning Robust Audio Representations From CLIP\", ICASSP 2022.
>
> [2] Fonseca et al., \"FSD50K: an Open Dataset of Human-Labeled Sound Events\", TASLPRO 2022.
>
> [3] Guzhov et al., \"AudioCLIP: Extending CLIP to Image, Text and Audio\", ICASSP 2022.

---

> ### Author Response · Authors · 2025-11-22
> **Author responses to Reviewer D4J1 (Part 2)**
>
> ## **W2: Regarding the time cost.**
>
> > *While exhibiting low memory consumption, DeepMORSE requires a slightly longer total training and testing time compared to baselines, such as TAC. Leading to challenges in real-world application deployment.*
>
> We appreciate the reviewer's comment on runtime and its impact on real-world deployment.
>
> ### **1) Training-time efficiency vs. TAC.**
>
> Although our DeepMORSE requires a slightly longer total training time than TAC, we argue that it is still efficient compared to TAC (Li et al., ICML 2024). In the revised version, we provide a plot of **clustering accuracy curve as a function of the training time** for both TAC and our DeepMORSE (see Fig. C.3), and explicitly mark the time point at which our DeepMORSE reaches TAC's best performance. As can be observed, to reach the same clustering accuracy as TAC, our DeepMORSE requires only about half the training time of TAC (and less than 30 seconds for each dataset). The longer total training time comes from our purpose to train our DeepMORSE to reach its own best performance, which is substantially higher than that of TAC. We therefore believe that the slightly longer total training time reflects a reasonable accuracy-efficiency trade-off. For practitioners who care about real-time performance, using early stopping at TAC-level accuracy will give our DeepMORSE a clear advantage. In addition, our DeepMORSE has **consistently lower memory consumption and per-epoch time** because TAC must load the neighbors of each sample together with the mini-batch. This makes DeepMORSE more favorable for deployment under constrained hardware.
>
> ### **2) Test-time cost and spectral clustering.**
>
> The increased test-time cost compared to TAC comes from the spectral clustering step, which requires solving an eigenproblem on the $N\times N$ affinity matrix constructed from the self-expressive coefficients. This is a standard bottleneck shared by most deep subspace clustering methods.
>
> Our focus in this work is on learning structured multimodal
> representations, so we follow the conventional subspace clustering
> pipeline and thus use spectral clustering without modification. However,
> there exist **scalable and differentiable spectral clustering variants**
> that could significantly reduce test-time overhead on very large
> datasets (e.g., Shaham et al., ICLR 2018; He et al., PR 2025). Developing a faster clustering head on top of our DeepMORSE's representations is an
> orthogonal but promising direction for future work.
>
> [1] Li et al., \"Image Clustering with External Guidance\", ICML 2024.
>
> [2] Uri Shaham, et al., \"SpectralNet: Spectral Clustering using Deep Neural
> Networks\", ICLR 2018.
>
> [3] Wei He, et al., \"Neural Normalized Cut: A Differential and
> Generalizable Approach for Spectral Clustering\", Pattern Recognition,
> Vol.164, 2025.

---

> ### Author Response · Authors · 2025-11-22
> **Author responses to Reviewer D4J1 (Part 3)**
>
> ## **W3: Regarding the "extra complexity vs. performance" trade-off.**
>
> > *The necessity of leveraging textual information necessitates a crucial pre-processing step, utilizing cross-modal sparse coding and a predefined dictionary to generate textual counterparts when image-text pairs are unavailable, thereby introducing external complexity and dependence on the quality and sparsity of this synthetic data. This unpaired data situation is very common in real-world settings. Furthermore, ablation studies reveal that DeepMORSE suffers a sharp degradation in clustering performance when components relying on the textual modality are removed, demonstrating that the overall effectiveness is highly dependent on the presence and successful integration of both modality expressions.*
>
> ### **1) Inherent trade-off.**
>
> From a high-level perspective, this reflects an inherent trade-off in
> representation learning: leveraging richer modalities requires some
> additional preprocessing, but in return yields better performance.
> Our DeepMORSE is not unique in this respect; most contemporary
> representation learning methods that improve over image-only baselines
> rely on the pretraining of large-scale multimodal data.
>
> ### **2) Complexity and robustness of textual counterpart generation.**
>
> -   This step is performed once offline, scales linearly with the number
>     of images, and does not affect the complexity of each training
>     iteration.
>
> -   Our experiments further validated that our DeepMORSE is robust to the
>     text generation method (as shown in Table 2 row 4) and the
>     hyper-parameters of the text generation (as shown in Fig. 4).
>
> ### **3) The performance drop when removing the text modality**
>
> Our DeepMORSE is explicitly designed to exploit additional semantic
> information from text. Thus, it is *expected* that removing the text
> modality leads to performance drop: the ablations (see rows 1 and 2 in
> Table 2) simply confirm that the model is indeed using the extra
> information **rather than ignoring it**.
>
> Overall, the observed improvements in clustering performance well support the
> necessity of incorporating text as an additional modality for image
> clustering. Although this requires a preprocessing step, the extra
> complexity is moderate, and our DeepMORSE remains relatively robust to
> variations in the quality of the generated textual counterparts.

---

> ### Author Response · Authors · 2025-11-22
> **Author responses to Reviewer D4J1 (Part 4)**
>
> ## **W4 / Q3: Regarding the hyper-parameter choice.**
>
> > *Although generally robust to hyperparameter choices, the model requires task-specific adjustments. Specifically, increasing the output dimensions and balancing the hyperparameter ($\gamma$) for downstream evaluations on datasets containing more categories suggests that the default configuration is not universally optimal for larger class counts.*
>
> > *The authors noted that for datasets with more than 128 categories (StanfordCars, SUN397), it was necessary to increase the output dimension $d$ and enlarge the balancing hyperparameter $\gamma$. Is there a principled rule or heuristic that can guide the selection of $d$ and $\gamma$ based on the number of classes $C$ or the complexity of the dataset to ensure the model maintains optimal performance and avoids convergence issues, instead of relying on manual tuning?*
>
> We appreciate the reviewer for this careful observation. We agree that a
> single "default" configuration cannot be universally optimal across different
> datasets which may contain very different numbers of classes. However, our
> adjustments follow a **simple and partially principled rule**, rather than a blindly ad-hoc
> tuning:
>
> 1.  **Adjusting the output dimension $d$.** Our DeepMORSE encourages the learned embeddings from different classes to occupy a set of orthogonal
>     subspaces. Thus, the output dimension must be greater than the
>     number of classes to accommodate these subspaces.
>
> 2.  **Adjusting the balancing hyper-parameter $\gamma$.** As justified in
>     (Meng et al., ICLR 2025), the upper bound of $\gamma$ scales
>     linearly with $\alpha=d/(N\epsilon^2)$, where $d$ is the output
>     dimension, $N$ is the batch-size, and $\epsilon$ is the coding
>     precision. Since that our DeepMORSE uses the same $N$ and $\epsilon$ for
>     all experiments, we have that $\gamma$ scales linearly with the output dimension
>     $d$. Empirically, our settings for $\gamma$ and $d$ approximately
>     follow $\gamma\approx0.5 \times d + 80$, which provides a simple
>     rule that can be used on other datasets without manual search for hyper-parameter.
>
> Overall, in our experiments, we have used the same hyper-parameters on 13 out of 15 datasets. For merely two datasets with 196 and 397 classes do we increase both $d$
> and $\gamma$ according to the principle above. Combined with our experiments to evaluate
> hyper-parameter sensitivity, we confirm that our DeepMORSE is generally robust to
> hyper-parameter choices, and only requires minimal, theoretically guided
> adjustment when the number of classes becomes very large. We thank the
> reviewer again for this careful observation and append these
> discussions in Appendix B.
>
> [1] Meng et al., \"Exploring a Principled Framework for Deep Subspace
> Clustering\", ICLR 2025.

---

> ### Author Response · Authors · 2025-11-22
> **Author responses to Reviewer D4J1 (Part 5)**
>
> ## **Q2: Regarding the theoretical analysis.**
>
> > *The paper explicitly lists the limitation that the theoretical underpinnings of modality-shared self-expression remain largely unexplored, leaving the fundamental working mechanism insufficiently understood. Can the authors elaborate on the specific challenges encountered when attempting to derive theoretical guarantees for the shared coefficient matrix $\boldsymbol{C}$ (Equation 4) in the multimodal setting, similar to those established for unimodal sparse subspace clustering? What are the most promising theoretical avenues for future research to address this gap?*
>
> We thank the reviewer for this thoughtful question. We agree that a full
> theoretical understanding of the modality-shared self-expression in Eq.
> (4) is largely unexplored and represents an important direction for
> future work. Here we sketch some challenges and possible avenues for the theoretical analysis to our DeepMORSE, in the hope that they may serve as a basis for future work.
>
> 1.  **From unimodal subspace clustering theory to the multimodal
>     scenario.**
>
>     Subspace clustering has a well-established theoretical foundation,
>     with representative works including Soltanolkotabi and Candés (AoS
>     2012), You and Vidal (ICML 2015), You et al. (ICCV 2019). These
>     analyses mainly focus on identifying geometric conditions under
>     which one can learn **subspace-preserving** self-expressive
>     coefficients, i.e., each data point is expressed only with points
>     from the same subspace. However, almost all of these theoretical results
>     are derived in the unimodal setting.
>
>     As an exception, we note that (Wang, IJCAI 2024) provides a necessary and sufficient
>     condition for **multi-view subspace-preserving recovery**, together
>     with a geometric characterization of the condition. In particular,
>     (Wang, IJCAI 2024) introduces multi-view formulations of incoherence
>     and circumradius, which can serve as a useful starting point for
>     theoretical analysis in the multimodal case. Nevertheless,
>     multimodal data typically involves much stronger cross-modal
>     discrepancies in the data distributions than multi-view scenarios.
>     Therefore, future work needs to explicitly address this additional
>     challenge.
>
> 2.  **From optimizing $\boldsymbol{C}$ to jointly optimizing
>     $\\{\boldsymbol{Z},\boldsymbol{C}\\}$.**
>
>     Previous theoretical analyses typically assume that data are fixed
>     and focus on analyzing the properties of the self-expressive matrix
>     $\boldsymbol{C}$. However, if incorporating representation learning,
>     the joint optimization over both $\\{\boldsymbol{Z},\boldsymbol{C}\\}$ makes
>     the analysis substantially more challenging.
>
>     Recently, we noticed an interesting work, (Meng et al., ICLR 2025), which shows that the deep self-expressive model
>     with a $\log\det(\cdot)$ regularizer promotes the learned representations (i.e., emdeddings) to lie
>     on a union of orthogonal subspaces. In the multimodal setting, we conjecture that this conclusion is likely to extend to each modality; nevertheless, the relations
>     between the subspace structures across different modalities remain unexplored.
>
> [1] Yulong Wang, \"Atomic Recovery Property for Multi-view Subspace-Preserving Recovery\", IJCAI 2024.
>
> [2] Soltanolkotabi and Candés, \"A geometric analysis of subspace clustering with outliers\", The Annals of Statistics, 2012.
>
> [3] You and Vidal, \"Geometric conditions for subspace-sparse recovery\", In ICML 2015.
>
> [4] You et al., \"Is an affine constraint needed for affine subspace clustering?\", In ICCV 2019.

---

> ### Author Response · Authors · 2025-11-22
> **Author responses to Reviewer D4J1 (Part 6)**
>
> ## **Q4: Regarding modal-independent $\boldsymbol{C}$.**
>
> > *Ablation studies show that DeepMORSE, which uses modality-shared self-expression yields significantly larger improvements than simply combining coefficients derived independently from each modality. Can the authors provide a more detailed, perhaps qualitative, explanation of why the rigid constraint of enforcing the exact same coefficient matrix $\boldsymbol{C}$ across both image and text representations (in Equation 4) is crucial for uncovering robust shared structures, compared to merely integrating two separate affinity matrices?*
>
> We thank the reviewer for this thoughtful question. We agree that the
> role of the modality-shared self-expressive matrix deserves a clearer
> explanation, and we provide both an intuitive hypothesis and empirical
> evidence.
>
> ### **1) Intuitive hypothesis:**
>
> Let $\boldsymbol{C}\_\text{img}$ and $\boldsymbol{C}\_\text{text}$ denote
> the self-expression matrices that are learned independently for the
> image and text modalities. In practice, both $\boldsymbol{C}\_\text{img}$
> and $\boldsymbol{C}\_\text{text}$ contain:
>
> - Correct coefficients, which correspond to samples from the same class,
>   and
>
> - Noise coefficients, which correspond to samples from different
>   classes.
>
> Intuitively, we expect that:
>
> - Correct relations tend to be shared across image and text.
>
> - Noise relations are not likely to be simultaneously supported in both
>   modalities.
>
> By enforcing a modality-shared self-expressive matrix, our DeepMORSE is
> encouraged to keep only those coefficients that are simultaneously
> useful for reconstructing both image and text representations, and to
> suppress coefficients that are only supported in one modality. As a
> result, the shared $\boldsymbol{C}$ tends to focus on coefficients that
> are more likely to be correct.
>
> This is exactly what we illustrate in Fig. 2: DeepMORSE retains only
> $c\_{2j}, c\_{3j}, c\_{4j}$, which are supported by both image and text,
> while discarding $c\_{1j}, c\_{5j}$, which appear only in one modality. In
> contrast, simply integrating two separate expressive coefficients (e.g., by
> averaging or max-pooling) cannot filter out the noise in each modality.
>
> ### **2) Empirical evidence:**
>
> To validate this hypothesis, We compare the shared matrix
> $\boldsymbol{C}$ learned by our DeepMORSE to those obtained by integrating
> $\boldsymbol{C}\_\text{img}$ and $\boldsymbol{C}\_\text{text}$. As shown
> in Figure C.2, the coefficient matrix from our DeepMORSE exhibits a much
> cleaner block-diagonal structure: off-block entries (inter-class
> relations) are strongly suppressed compared to the integrated version.
>
> Quantitatively, for each method, we compute (over 10 trials) the average
> absolute value of block-diagonal entries (for intra-class relations), and the average
> absolute value of off-block-diagonal entries (for inter-class relations). The results
> show that:
>
> - The inter-class (noise) relations of our DeepMORSE are consistently
>   smaller than those of the integrated-coefficients baseline.
>
> - The intra-class relations of our DeepMORSE are comparable to those of the
>   baseline..
>
> These observations support our hypothesis that modality-shared
> coefficient encourages our DeepMORSE to learn robust relations among data
> and yields a "cleaner" affinity for clustering. This cleaner affinity
> directly contributes to the performance improvement gap observed in the ablation
> studies.

---

> ### Author Response · Authors · 2025-12-02
>
> We appreciate the insightful comments and suggestions from reviewer D4J1, and kindly invite the reviewer to take some time to read our responses in the rebuttal to see if there are still unclear issues to clarify. Thanks!

---

### Official Review · Reviewer_JHm3 · 2025-11-02

**Soundness:** 3
**Presentation:** 3
**Contribution:** 3
**Rating:** 6
**Confidence:** 4

**Summary:**

This paper aims to leverage textual information for image clustering. Specifically, this work assumes that there is a modality-invariant relationship within both the vision and text subspaces, i.e., each data point can be linearly represented by the same set of other data points using the same coefficients for both vision and text after appropriate transformations. Thereafter, enhanced vision representations can be learned and regularized by the textual information. For each image, the textual information is learned by a sparse coding using the dictionary to cover the image's representation. The proposed method shows better performance on various datasets. Moreover, the contribution of each component from the proposed method is well demonstrated and the proposed combination shows the best performance.

**Strengths:**

1) To leverage the textual information for clustering images, this work proposes to learn enhanced image representations constrained by the modality-invariant relationship between data points. Specially, after appropriate transformations, each data point can be represented by a linear combinations of other datapoints using the same coefficients for both the vision and textual representations. The proposed optimization framework is sound accordingly.

2) Compared to the reported baselines, the proposed method provides better performance on various datasets, which demonstrate the effectiveness of the proposal. Moreover, the ablation study shows the contribution of each component well.

3) The paper is well written motivated by sound discussion and easy to follow.

**Weaknesses:**

1) The proposed method employs a set of transformations f, g for each modality. It would be interesting to show that this transformation is necessary through the experiments. For example, how about the performance without doing the transformation, while keep all other learning objectives.

2) Given the vision and text representations, it is applicable to treat each as a view for each data point. Showing the state-of-the-art multi-view clustering using both of them would help sufficiently demonstrate that multi-view clustering is not that helpful compared to the proposed method.

3) Some strong unimodal deep clustering methods are discussed or compared, e.g., CoKe, SeCu. Any reasons for that?

**Questions:**

Related questions can be found in the weakness section.

---

> ### Author Response · Authors · 2025-11-22
> **Author responses to Reviewer JHm3 (Part 1)**
>
> We appreciate the reviewer JHm3 for reading our submission carefully, for the supportive rating, and for recognizing our paper as "well written motivated by sound discussion and easy to follow".
>
> ## **W1: Regarding the transformations $f$ and $g$.**
>
> > *The proposed method employs a set of transformations f, g for each modality. It would be interesting to show that this transformation is necessary through the experiments. For example, how about the performance without doing the transformation, while keep all other learning objectives.*
>
> We appreciate the reviewer's suggestion to empirically verify the
> necessity of using the transformations $f$ and $g$. Intuitively, these transformations serve two purposes:
>
> 1.  **Linearization**: The $f$ and $g$ transform the original inputs (which potentially lie on a union of nonlinear manifolds) to the target space where the distribution of the embeddings is well approximated by a union of linear subspaces;
>
> 2.  **Separation**: The transformations $f$ and $g$ are able to amend the subspaces, in which the embeddings lie, to be more ``separable'' (ideally tending towards orthogonal) and thus facilitate the self-expressive modeling and subsequent clustering.
>
> To validate the necessity and the effect of these transformations, we conduct an ablation study with three variants:
>
> 1.  **Without $f$ and $g$**: the self-expressive model is learned directly on the frozen CLIP features;
>
> 2.  **Linear $f$ and $g$**: $f$ and $g$ are replaced by learnable linear projections;
>
> 3.  **Nonlinear $f$ and $g$ (DeepMORSE)**: our full model with deep nonlinear transformations.
>
> For each variant, we report the average clustering accuracy over 5 runs with the best-tuned hyper-parameters in the following table.
>
> | Variant                | CIFAR-10 | CIFAR-20 | Dogs-15  | DTD-47   | UCF-101  |
> |------------------------|---------:|---------:|---------:|---------:|---------:|
> | w/o $f$ and $g$              | 78.2±0.2 | 53.0±1.0 | 61.7±1.9 | 45.8±0.9 | 60.3±0.4 |
> | w/ linear $f$ and $g$        | 92.1±0.1 | 59.2±0.9 | 82.3±0.6 | 54.4±0.3 | 70.8±0.5 |
> | DeepMORSE              | **92.3**±0.1 | **63.3**±0.6 | **88.7**±0.5 | **55.0**±0.4 | **73.2**±0.5 |
>
> As can be observed, we have the following conclusions:
>
> 1.  **Without using $f$ and $g$**, the performance drops noticeably compared to our DeepMORSE. This confirms that the transformations $f$ and $g$ are indeed necessary. Interestingly, this variant still outperforms "CLIP ($k$-means)", indicating that the raw features are more likely to be approximated by a union of subspaces than by a set of centroids.
>
> 2.  **With linear $f$ and $g$**, the performance improves substantially over the "w/o $f$ and $g$" case, suggesting that even using a linear projection can help align the data towards a union-of-subspaces structure and thus enhance the clustering performance.
>
> 3.  **With nonlinear $f$ and $g$ (i.e., our DeepMORSE)**, we obtain the best performance across all datasets. This suggests that our original design choice of using deep nonlinear transformations is more likely able to linearize the input data and amend the subspaces of the embeddings more "separable" (i.e., with lower coherence).
>
> We thank the reviewer again for this insightful suggestion and have added these results and the discussions to Appendix C of the revised submission.

---

> ### Author Response · Authors · 2025-11-22
> **Author responses to Reviewer JHm3 (Part 2)**
>
> ## **W2: Regarding multi-view clustering methods.**
>
> > *Given the vision and text representations, it is applicable to treat each as a view for each data point. Showing the state-of-the-art multi-view clustering using both of them would help sufficiently demonstrate that multi-view clustering is not that helpful compared to the proposed method.*
>
> We thank the reviewer for the insightful suggestion. Indeed, the CLIP image and text embeddings can naturally be regarded as two views of each data point. To thoroughly evaluate our DeepMORSE in this context, we conduct the following experiments.
>
> ### **1) State-of-the-art multi-view clustering on our multimodal datasets.**
>
> We first follow the reviewer's suggestion and treat the CLIP image and text embeddings as two views, respectively, and then apply state-of-the-art multi-view clustering methods, e.g., CANDY (Guo et al., NeurIPS 2024) and COPER (Eisenberg et al., ICLR 2025), to our multimodal data (after text generation). For fairness, we keep network architectures the same as in our DeepMORSE. All experiments are repeated with 5 random seeds, and we report the average performance (ACC\% / NMI\% / ARI\%) in the table below.
>
> | Method                           | ImageNet-Dogs | DTD | UCF101 |
> |----------------------------------|------------------:|--------:|-----------:|
> | CANDY (Guo et al., NeurIPS 2024) | 81.6 / 80.4 / 71.1             | 51.5 / 62.5 / 35.5    | 56.2 / 77.0 / 47.4       |
> |COPER (Eisenberg et al., ICLR 2025) |  82.0 / 80.8 / 72.2  | 50.1 / 61.3 / 33.8  |  56.5 / 77.8 / 47.6 |
> | Ours                             | **88.7** / **84.4** / **78.3**         | **55.0** / **64.9** / **39.3**| **73.2** / **86.1** / **66.7**   |
>
>
> Across all three datasets, our DeepMORSE consistently outperforms the multi-view baselines, implying that modeling modality-specific UoS structure with modality-shared relationship, as done in DeepMORSE, provides a more effective way to leverage multimodal information.
>
> ### **2) DeepMORSE on standard multi-view benchmarks.**
>
> We then evaluate our DeepMORSE on several standard multi-view clustering datasets, treating their multiple views as separate modalities in our framework. As shown below (ACC\% / NMI\% / ARI\%), our DeepMORSE is competitive to, and often
> outperforms, the strong multi-view clustering methods.
>
>
> | Method                           | Scene15  | LandUse21  | NUS-WIDE  |
> |----------------------------------|-----------:|-----------------:|----------:|
> | CANDY (Guo et al., NeurIPS 2024) | 42.0 / 41.6 / 24.7        | 30.6 / 36.5 / 16.2          | 62.1 / 49.0 / 37.0         |
> |COPER (Eisenberg et al., ICLR 2025) | 40.7 / 42.0 / 25.0 | 31.0 / 35.9 / 16.1 | 62.1 / 49.3 / 37.7  |
> | Ours                             | **43.2** / **46.2** / **28.0**    | **31.4** / **37.9** / **17.4**      | **63.4** / **50.6** / **43.2**     |
>
>
> We thank the reviewer again for the insightful suggestion and have
> included these results in the revised manuscript (Appendix C).
>
> [1] Guo et al., \"Robust Contrastive Multi-view Clustering against Dual Noisy Correspondence\", NeurIPS 2024.
>
> [2] Eisenberg et al., \"COPER: Correlation-based Permutations for Multi-View Clustering\", ICLR 2025.

---

> ### Author Response · Authors · 2025-11-22
> **Author responses to Reviewer JHm3 (Part 3)**
>
> ## **W3: Regarding CoKe and SeCu.**
>
> > *Some strong unimodal deep clustering methods are discussed or compared, e.g., CoKe, SeCu. Any reasons for that?*
>
> We thank the reviewer for clearly highlighting CoKe and SeCu as strong unimodal deep clustering approaches. In our submission, we did not directly compare our DeepMORSE to CoKe and SeCu, because a fair comparison is difficult to conduct: a) these unimodal methods do not naturally extend to the multimodal setting; and b) our DeepMORSE does not benefit from the data augmentations on which both CoKe and SeCu rely. However, we agree with the reviewer that they are highly relevant prior works and thus we have expanded both our discussions and our comparison with them in the revised manuscript.
>
> Note that CoKe, SeCu, and our DeepMORSE all aim to improve the quality of learned representations for deep clustering by designing novel training objectives: CoKe introduces a metric-learning-style pretext task combined with an online constrained $k$-means; SeCu effectively eliminates the direction of negative instances from the gradient for stable training; our DeepMORSE introduces a deep modality-shared self-expressive model that encourages multimodal representations to follow a  modality-specific union-of-subspaces structure, coupled via a shared coefficient matrix.
>
> Despite the shared high-level goal, there are important differences:
>
> - **Representation geometry.** CoKe and SeCu ultimately model each
>   cluster by a centroid in the embedding space (which can be viewed as a zero-dimensional subspace). In contrast, our DeepMORSE explicitly models clusters as low-dimensional subspaces and the distribution of the data as a union of subspaces.
>
> - **Modalities.** CoKe and SeCu are unimodal image clustering methods and, to the best of our knowledge, have not yet been extended to multimodal settings. Our DeepMORSE, by design, targets multimodal data.
>
> At the same time, we believe that the mechanisms in CoKe and SeCu are interesting for extending our framework. For example, it is appealing to incorporate the cluster-size (CoKe) or entropy-style (SeCu) constraints into our DeepMORSE as an alternative solution to prevent collapse issue, alongside our current $\log\det(\cdot)$ regularizer. We leave it for future work.
>
> [1] Qi Qian, \"Stable Cluster Discrimination for Deep Clustering\", ICCV 2023.
>
> [2] Qian et al., \"Unsupervised Visual Representation Learning by Online Constrained K-Means\", CVPR 2022.

---

> ### Author Response · Authors · 2025-12-02
>
> We thank the reviewer JHm3 again for this insightful suggestions and kindly invite the reviewer to take a few minutes to read our responses to see if there are other issues to clarifiy. Thanks!

---

### Author Response · Authors · 2025-11-22
**Summary of our revisions**

We would like to thank all reviewers for the time and effort in reviewing our paper.
To address reviewers' comments and concerns, we have made the following changes.

-  We have conducted experiments to verify the necessity
of involving transformations $f$ and $g$ in Table C.4 of **Appendix C**. (**JHm3**)
-  We have conducted experiments to evaluate our DeepMORSE in a multi-view clustering setting on both our benchmarks and multi-view datasets in Table C.5 of **Appendix C**. (**JHm3**)
-  We have added a discussion and compared with additional baselines, e.g., CoKe and SeCu (L121-126; Table 1). (**JHm3**)
-  We have conducted experiments to extend our DeepMORSE to the acoustic modality in Table C.8 of **Appendix C**. (**D4J1** and **4ETM**)
-  We have added a detailed time cost comparison to the baseline in Figure C.3 of **Appendix C**. (**D4J1**)
-  We have added a simple and partially principled rule for hyper-parameter setting (L955-965). (**D4J1** and **4ETM**)
-  We have conducted experiments to clarify why enforcing modality-shared coefficients is beneficial in Figure C.2 of **Appendix C**. (**D4J1**)
-  We have refined the wording and strengthened the empirical evidence supporting the motivation (L76-80; Fig.1; Fig.5). (**xkPq**)
-  We have clarified the experimental settings of CLIP ($k$-means) and PRO-DSC in Table 1, as well as the configuration used in the second row of the ablation study (L351-352; L974-987). (**xkPq**)
-  We have extended the sensitivity study to all image clustering benchmarks in Figure 4. (**xkPq**)
-  We have evaluated our DeepMORSE using MLLM-generated captions as text counterparts in Table C.3 of **Appendix C**. (**xkPq**)

The revised contents are highlighted in blue in the revised paper. We kindly invite all reviewers to take a look at these updated results and our responses.

We sincerely thank all reviewers again for their valuable suggestions, which have greatly helped improve the quality of our work. If you have any further questions, we would be very happy to discuss them further.

---

### Meta-Review · Area_Chair_Pi6d · 2026-01-06

**Summary:**

Reviewer JHm3: This work proposes to learn enhanced image representations constrained by the modality-invariant relationship between data points. The proposed method provides better performance on various datasets, which demonstrate the effectiveness of the proposal. The paper is well written motivated by sound discussion and easy to follow.
. However, the reviewer still has some concerns on the weaknesses about the lack of more validated experiments, unclear discussion in the experiments.

 Reviewer D4J1: This paper's core strength lies in its simplicity. DeepMORSE overcomes limitations of prior alignment methods by simultaneously learning structured representations. The approach achieves descent clustering performance across seven benchmarks. Furthermore, the learned structured representations demonstrate transferability and robustness.  However, the reviewer still has some concerns on the weaknesses about the limitation in scope to vision-language data, and the necessary extension of the method to other modalities, lack of more running time analysis, lack of task-specific adjustments.


 Reviewer 4ETM: Experimental results show consistent improvements over strong baselines across multiple datasets. The paper clearly identifies the limitations of enforcing overly tight shared embedding spaces. The motivation for learning modality-specific representations is intuitive.  However, the reviewer still has some concerns on the weaknesses about the lack of deeper theoretical justification for the chosen balance, unclear contributions of each component of the model, lack of discussion of the efficiency trade-off.

Reviewer xkPq: This paper proposes a simple-yet-effective method for multi-modal image clustering. The proposed model can jointly learn representations conforming to a union of modality-specific subspaces and discovers shared structures across modalities. The learned structured representations can be directly applied to downstream tasks including image retrieval and zero-shot classification. However, the reviewer still has some concerns on the weaknesses about clarification on the motivation, lack of several details of the method details, insufficient hyperparameter analysis, lack of more experiments.

**Reviewer Concerns:**

After carefully evaluating the rebuttals, I think the reviews from the Reviewer JHm3 were addressed from the response.
For the remaining reviewer concerns, they are all not fully addressed.

**Reviewer Scores:**

For the Reviewer D4J1, 4ETM,  xkPq, I think the reviewer may keep the rating unchanged based on the response.

For the  Reviewer  JHm3 , I think the reviewer may increase the rating or keep the rating unchanged based on the response.

---

### Decision · Program_Chairs · 2026-01-26

Reject